# Capturing Gaze Shifts for Guidance:
# Cross-Modal Fusion Enhancement for VLM Hallucination Mitigation

**Zheng Qi** [1]   **Chao Shang** [1]   **Evangelia Spiliopoulou** [1]   **Nikolaos Pappas** [2] [†]

## Abstract

Vision language models (VLMs) often generate hallucination, i.e., content that cannot be substantiated by either textual or visual inputs. Prior work primarily attributes this to over-reliance on linguistic prior knowledge rather than visual inputs. Some methods attempt to mitigate hallucination by amplifying visual token attention proportionally to their attention scores. However, these methods overlook the visual attention sink problem, where attention is frequently misallocated to task-irrelevant visual regions, and neglect cross-modal fusion balance by enhancing only visual attention without adjusting attention to the user query. This can result in amplifying incorrect areas while failing to properly interpret the user query. To address these challenges, we propose a simple yet effective method called **G**aze Sh**i**ft-Guided Cross-modal **F**usion Enhancemen**t** (**GIFT**). GIFT pre-computes a holistic visual saliency map by tracking positive changes in visual attention, or *"gaze shifts"*, during user query comprehension, and leverages this map to amplify attention to both salient visual information and the user query at each decoding step. This reduces the impact of visual attention sink, as irrelevant tokens exhibit minimal shifts, while ensuring balanced cross-modal fusion for well-integrated representation. Extensive experiments show that GIFT effectively mitigates hallucination in VLMs across both generative and classification tasks, achieving up to 20.7% improvement over greedy decoding, while maintaining general vision-language performance with low computational overhead.

## 1. Introduction

Vision language models (VLMs) (Li et al., 2023c; Liu et al., 2023b; Zhu et al., 2023; Liu et al., 2024a; Hurst et al., 2024; Wang et al., 2024b; Bai et al., 2025) have achieved remarkable progress on tasks that require joint reasoning over textual and visual information, such as visual question answering, visual reasoning, and image captioning. Despite these advances, VLMs remain prone to generating hallucination, i.e., content that cannot be substantiated by either textual or visual inputs (Liu et al., 2024b). This issue poses serious challenges, particularly in high-stakes domains such as biomedicine (Li et al., 2023b; Chen et al., 2024b), autonomous driving (Wang et al., 2023b; Li et al., 2025), and robotics (Chen et al., 2024a; Li et al., 2024), where factual accuracy is critical for safe and effective operation.

Recent analyses suggest these failures stem from vision language models (VLMs) over-relying on linguistic prior knowledge while under-utilizing visual inputs (Wang et al., 2024a; Zhang et al., 2024). To mitigate this, inference-time interventions have been proposed to enhance visual grounding by highlighting visual signals based on visual saliency, i.e., the relevance of specific visual regions to the task at hand. For instance, Yin et al. (2025) enhances attention allocated to visual tokens during decoding in proportion to their attention scores. While this approach strengthens visual contribution, it does not account for the balance between visual and query signals during cross-modal fusion. Consequently, the model may attend to relevant regions but misinterpret the query, forming inaccurate integrated representations. Moreover, this approach does not address the issue of visual attention sink (Kang et al., 2025), where attention is persistently misallocated to irrelevant visual tokens, potentially amplifying incorrect regions throughout generation. Existing methods recalibrate visual token attention to mitigate this issue (Kang et al., 2025; Zhu et al., 2025b), but fail to address the broader problem of insufficient overall visual contribution during decoding.

To address these limitations, we propose **G**aze Sh**i**ft-Guided Cross-modal **F**usion Enhancemen**t** (**GIFT**), an inference-time hallucination mitigation method for VLMs. Drawing inspiration from human vision, we hypothesize that VLMs, like humans, dynamically shift their "visual gaze" when

---

[†]Work done while at Amazon. [1]AWS AI Labs [2]Oracle AI. Correspondence to: Zheng Qi <zhengqii@amazon.com>.

*Proceedings of the $43^{rd}$ International Conference on Machine Learning*, Seoul, South Korea. PMLR 306, 2026. Copyright 2026 by the author(s).

processing information-rich words in a user query. By tracking positive changes in visual token attention, i.e., "gaze shifts", over these information-rich query tokens at the layer exhibiting the largest positive change, GIFT pre-computes a holistic visual saliency map that captures task-relevant regions prior to decoding, requiring pre-filling only up to that layer. This mechanism also mitigates the impact of visual attention sink, as irrelevant regions exhibit minimal or no attention shift. During decoding, GIFT leverages this saliency map to proportionally amplify attention to salient visual tokens in critical cross-modal fusion layers, where the model attends strongly to both visual and query tokens. Unlike Yin et al. (2025), which only increases visual token attention, GIFT also adjusts query token attention based on the overall visual attention amplification ratio, maintaining cross-modal balance and forming well-integrated representations.

In summary, our main contributions are three-fold:

- We introduce a novel mechanism that captures a holistic view of salient visual regions while effectively mitigating the visual attention sink problem. This mechanism pre-computes a task-relevant visual saliency map prior to decoding by tracking positive shifts in visual attention, i.e., "gaze shifts", as the VLM processes information-rich words in a user query.

- We propose GIFT, a lightweight inference-time hallucination mitigation method that leverages the precomputed saliency map to guide visual attention enhancement while proportionally scaling attention to query tokens to preserve cross-modal fusion balance.

- We demonstrate that GIFT consistently mitigates hallucination across VLM architectures and model sizes, achieving gains of up to 20.7% on CHAIR, 15.9% on MMHal-Bench, and 3.0% on POPE, while preserving general vision-language performance with low computational overhead. Extensive ablation studies validate component contributions and robustness to hyperparameter selection.

## 2. Related Work

**VLM Hallucination Mitigation.** A key cause of hallucination in VLMs is over-reliance on linguistic prior knowledge rather than visual inputs (Wang et al., 2024a; Zhang et al., 2024). Training-based approaches address this by introducing specialized learnable modules (Zhao et al., 2024) or curated data augmentations (Liu et al., 2023a; Pi et al., 2024; Chen et al., 2025; Sarkar et al., 2025; Wu et al., 2026) to encourage stronger reliance on visual features. While effective, these methods incur high computational costs and limited scalability.

Inference-time mitigation methods offer a more flexible alternative and can be broadly categorized into three types: (1)

*Contrastive Decoding* (Leng et al., 2024; Liu et al., 2024c; Huo et al., 2024; Wang et al., 2025; Zhu et al., 2025a) reduces over-reliance on knowledge priors by contrasting output distributions of two inputs, one with original visual inputs and one with perturbed or absent visual inputs. However, this approach requires generating counterpart outputs, incurring significant computational overhead. Our method instead strengthens visual contributions directly in intermediate layers, eliminating the need for generating alternatives. (2) *Visual Input Modification* manipulates the raw image to emphasize salient regions derived from intermediate signals, by blurring irrelevant areas (Yu et al., 2024), magnifying key regions (Mao et al., 2025), or cropping salient patches (Zhang et al., 2025). These techniques typically require additional forward passes or auxiliary inputs, increasing computational cost, and struggle when multiple regions are salient or when a single region is overly large. In contrast, our method operates directly on intermediate outputs without constraints on the number or size of relevant regions. (3) *Attention Steering* directly steers attention towards visual tokens, either by applying a constant value (Zhu et al., 2025a) or scaling proportionally to attention scores (Yin et al., 2025). While these methods increase visual contribution, they often neglect cross-modal fusion balance, i.e., overemphasizing visual features without adequately reinforcing query token attention can impair comprehension. They also overlook the visual attention sink problem (Kang et al., 2025), which our approach explicitly addresses.

**Attention Sink.** Attention sink refers to the phenomenon where task-irrelevant tokens, such as those with limited semantic meaning or representing background, receive disproportionately high attention weights. This issue has been explored in language models (Xiao et al., 2024; Ferrando & Voita, 2024; Qiu et al., 2026), vision transformers (Darcet et al., 2024), and VLMs (Kang et al., 2025; Zhu et al., 2025b). While these VLM mitigation strategies recalibrate visual attention to suppress sink tokens, they do not address the broader limitation that visual features contribute insufficiently during generation. In this work, we present a simple yet effective method that pre-computes a task-relevant visual saliency map by tracking positive changes in visual attention, i.e., "gaze shifts", over information-rich query tokens. During decoding, this saliency map, which is robust to visual attention sink, guides joint amplification of attention to both visual and query tokens, improving cross-modal integration.

## 3. Visual Saliency Map Computation via Gaze Shift Tracking

We present our mechanism for computing a visual saliency map that captures salient visual regions while mitigating the visual attention sink problem. We first examine whether

Q: *Is there a banana in the image?* A: *Yes.*

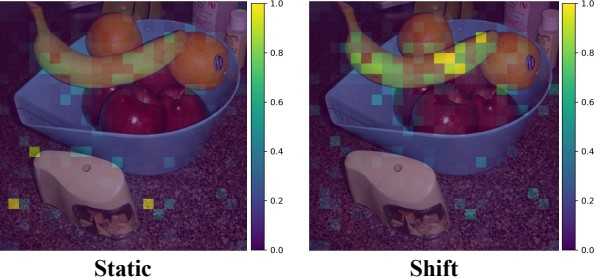

Q: *Is this a Macbook or Windows laptop?* A: *Macbook, as indicated by the presence of the Apple logo.*

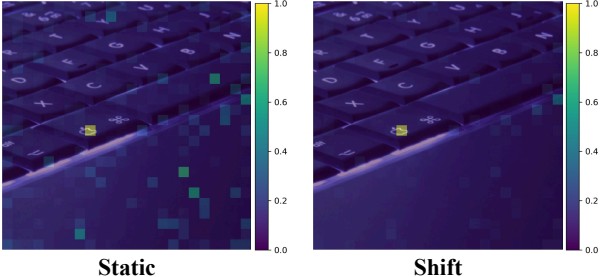

Q: *Is there a motorcycle in the image?* A: *No.*

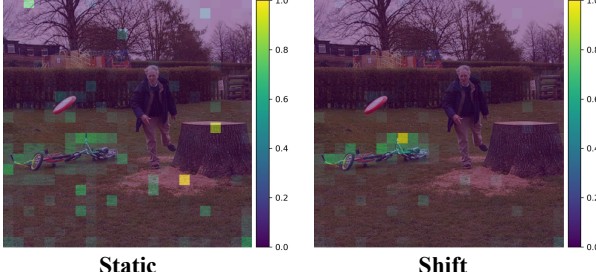

Q: *Which cat (left, right or middle) opens its mouth?* A: *The cat on the right side opens its mouth.*

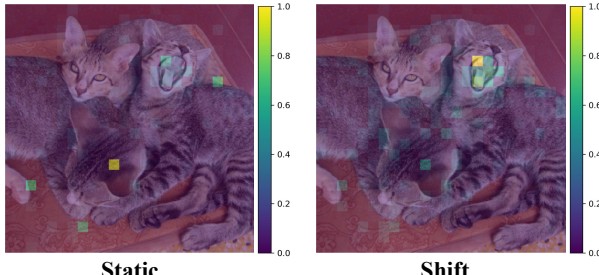

*Figure 1.* **Comparison of visual saliency maps from LLaVA 1.5 7B.** The vanilla method (**Static Gaze**) averages visual token attention over all query tokens, while the proposed method (**Gaze Shift**) averages positive changes in visual token attention over information-rich query tokens. **Gaze Shift** more effectively highlights task-relevant visual regions and reduces noise from visual attention sink.

a simple average of visual attention across user query tokens, referred to as **"static gaze"**, can effectively produce a holistic, noise-free visual saliency map.

Vision language models (VLMs) typically process three inputs: a system instruction $s$, visual inputs $v$, and a user text query $t$. The system instruction and query are tokenized into sequences $X_S$ and $X_T$, while the visual input $v$ is encoded by a visual encoder into dense embeddings and then projected into text-aligned visual tokens $X_V$ (Liu et al., 2023b; Wang et al., 2024b). These components are concatenated as $X = [X_S; X_V; X_T]$, and passed into a large language model (LLM) to generate output tokens autoregressively:

$$y_t = \arg\max p_\theta \left(y_t \mid y_{<t}, X_S, X_V, X_T\right), \qquad (1)$$

where $y_{<t}$ denotes the previously generated tokens.

Within the model, the attention matrix $\boldsymbol{A}^l \in \mathbb{R}^{h \times n \times n}$ encodes how each of the $n$ tokens attends to all others across $h$ attention heads at layer $l$. For simplicity, batch dimensions are omitted. Visual attention sink (Kang et al., 2025) refers to the phenomenon where query-irrelevant visual tokens receive disproportionately high attention. From this matrix, we extract the submatrix representing attention from query tokens $X_T$ to visual tokens $X_V$, which serves as the foundation for constructing the visual saliency map.

Prior work (He et al., 2024; Yin et al., 2025; Kang et al.,

2025) has demonstrated that only a subset of attention heads primarily attend to visual information. Following their findings, at each layer $l$, we select the top 50% of attention heads with the highest cumulative attention to visual tokens $X_V$ aggregated across all query tokens $X_T$, denoted as $\mathcal{H}_{TV}^l$. We then compute the mean attention over these selected heads and average across query tokens to produce the saliency map $\mathcal{S}^l$:

$$\mathcal{S}^l = \frac{1}{|\mathcal{H}_{TV}^l| \cdot |X_T|} \sum_{h \in \mathcal{H}_{TV}^l} \sum_{i \in X_T} \boldsymbol{A}_{h,i,j}^l, \qquad (2)$$

where $h$ indexes attention heads $\mathcal{H}_{TV}^l$, $i$ indexes query tokens $X_T$, and $j$ indexes visual tokens $X_V$. We apply min-max normalization to scale the saliency map to $[0, 1]$.

Figure 1 shows "static" saliency maps from LLaVA-1.5 7B (Liu et al., 2023b) on the left side of each example. While they partially highlight relevant visual regions, they often assign high saliency scores to irrelevant areas as well. This misallocation, known as visual attention sink (Kang et al., 2025), produces misleading signals that can negatively affect downstream generation.

To address this, we propose a simple yet effective approach inspired by human vision. We hypothesize that, like humans, VLMs dynamically shift their "visual gaze" to capture relevant visual information while comprehending the user query.

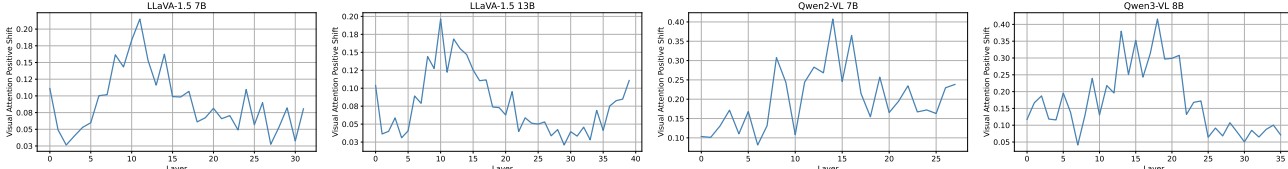

*Figure 2.* **Volume of positive visual attention shifts in VLMs when processing information-rich query tokens across layers.** The volume reflects how strongly the model reallocates focus within visual regions, with the largest shifts occurring in early to middle layers, indicating that VLMs settle on relevant visual regions once identified.

By tracking positive changes in visual attention, referred to as **"gaze shifts"**, over information-rich query tokens, we obtain a holistic view of task-relevant visual regions. Since irrelevant regions typically exhibit minimal or no change in attention, this approach naturally mitigates the issue of visual attention sink. We restrict tracking to information-rich words, where attention shifts are most meaningful, and consider only positive shifts, since negative shifts merely indicate moving focus away from previously salient regions and would cancel out meaningful increases.

Concretely, we first extract information-rich words from the user query using spaCy's[1] Part-Of-Speech (POS) tagging with minimal computational overhead, selecting words tagged as NOUN, PROPN, VERB, ADJ, ADV, or NUM. These words correspond to a set of query tokens $X_{Tr}$. At layer $l$, we select the top 50% of attention heads, $\hat{\mathcal{H}}_{TrV}^l$, with the highest cumulative positive changes in attention to visual tokens aggregated across these information-rich query tokens. Here, positive change in attention is defined as the increase in visual attention from the previous query to the current one, with negative changes set to zero. Using these heads and query tokens, we compute a refined saliency map that captures the average positive shift in visual attention, emphasizing task-relevant regions:

$$\Delta \mathbf{A}_{h,i,j}^l = \max(\mathbf{A}_{h,i,j}^l - \mathbf{A}_{h,i-1,j}^l, 0)$$
$$\hat{\mathcal{S}}^l = \frac{1}{|\hat{\mathcal{H}}_{TrV}^l| \cdot |X_{Tr}|} \sum_{h \in \hat{\mathcal{H}}_{TrV}^l} \sum_{i \in X_{Tr}} \Delta \mathbf{A}_{h,i,j}^l. \quad (3)$$

As with "static" saliency maps, we apply min-max normalization to scale the saliency map to $[0, 1]$.

Figure 1 shows the resulting "shift" saliency maps from LLaVA-1.5 7B on the right side of each example, which more accurately highlight relevant regions and reduce noise from irrelevant areas compared to "static" maps. Gaze shifts also adapt to query intent even when the query does not explicitly name target objects: they spread across all visually grounded elements for open-ended queries such as image captioning, and follow background cues when the query

---

[1] https://spacy.io

*Table 1.* **Comparison of visual saliency computation methods.** Scores measure saliency concentration within bounding boxes, normalized by box area. Higher is better.

| Method | Static | Shift |
|---|---|---|
| Norm. Saliency Score | 5.40 | **11.92** |

references abstract scene properties such as weather. We provide examples for both cases in Appendix H.

To select the optimal layer for computing visual saliency maps, we identify where visual attention is most dynamically realigned during query processing by selecting the layer with the largest sum of $\hat{\mathcal{S}}^l$ without min-max normalization. Figure 2 shows that, on 50 examples sampled from the training set of TextVQA (Singh et al., 2019), a visual question answering dataset, this peak occurs in early to middle layers across models. We argue that the location of this peak is an intrinsic property of each model rather than a dataset-specific artifact. To support this, we additionally measure the quantity on MathVista (Lu et al., 2024), a math reasoning dataset in visual contexts that differs substantially in task type. For both LLaVA-1.5 7B and Qwen3-VL 8B, the peak layer coincides with that on TextVQA and remains stable across random samples on both datasets. Detailed per-layer values across random samples are reported in Appendix A. In the following sections, we denote the visual saliency map extracted from this layer as $\hat{\mathcal{S}}$.

To validate that gaze shift tracking produces more accurate saliency maps, we quantitatively evaluate both approaches on their ability to localize task-relevant visual regions. We use 1,000 examples from the MSCOCO 2014 training set (Lin et al., 2014), each containing an image with an object instance and its bounding box, restricting to instances whose category appears only once to avoid ambiguity. For each example, we query the model with "Is there a {object} in the image?" and extract saliency maps using both methods. We normalize each saliency map to sum to 1 and compute a normalized saliency score: the proportion of saliency within the bounding box divided by the box's relative area. This metric measures how well the saliency map concentrates on the relevant object while accounting for box size. Table 1 shows that the "shift" method achieves more than

double the score of "static" (11.92 vs. 5.40), demonstrating significantly stronger focus on task-relevant regions.

# 4. Gaze Shift-Guided Cross-modal Fusion Enhancement

We now introduce our hallucination mitigation method, illustrated in Figure 3, which leverages the saliency map to guide cross-modal fusion enhancement through attention steering during decoding.

**Selecting Enhancement Layers.** We first identify which layers are most effective for enhancing cross-modal fusion. Unlike the visual saliency map in Section 3, which tracks attention flow from query tokens to visual tokens, here we analyze flow from output tokens to query and visual tokens, respectively. Using the 50 TextVQA examples, we measure the proportion of attention allocated to visual tokens and query tokens over information-rich output words $Y_r$, denoted as $\mathcal{R}_V^l$ and $\mathcal{R}_T^l$. At each layer, we retain the top 50% attention heads with the highest values, denoted as $\mathcal{H}_{OV}^l$ and $\mathcal{H}_{OT}^l$. To ensure attention patterns are not dominated by linguistic priors, which intensify with output length (Min et al., 2024; Xie et al., 2025), we restrict outputs to a single sentence. Formally:

$$\mathcal{R}_V^l = \frac{1}{|\mathcal{H}_{OV}^l| \cdot |Y_r|} \sum_{h \in \mathcal{H}_{OV}^l} \sum_{i \in Y_r} \sum_{j \in X_V} \boldsymbol{A}_{h,i,j}^l$$
$$\mathcal{R}_T^l = \frac{1}{|\mathcal{H}_{OT}^l| \cdot |Y_r|} \sum_{h \in \mathcal{H}_{OT}^l} \sum_{i \in Y_r} \sum_{j \in X_T} \boldsymbol{A}_{h,i,j}^l. \quad (4)$$

Figure 4 shows that across models, attention to visual and query tokens exhibits broadly aligned layer-wise trends, particularly in middle layers, highlighting their joint contribution to well-integrated representations, though absolute attention levels remain low. This indicates that effective cross-modal fusion requires simultaneously enhancing attention to both modalities. We therefore select layers with high attention to both visual and query tokens for enhancement, denoted as $\mathcal{L}$, as these are where cross-modal fusion is most active. We detail the layer selection process in Appendix D and demonstrate that our method is robust to layer selection in Section 5.3.

**Enhancing Visual Attention via Attention Steering.** Attention steering (Zhang et al., 2023) aims to bias the attention matrix $\boldsymbol{A}$ toward salient tokens by adding a learned or heuristic bias $\boldsymbol{B}$:

$$\boldsymbol{A} = \text{Softmax}\left( \frac{\boldsymbol{Q} \cdot \boldsymbol{K}^\top}{\sqrt{d_k}} + \boldsymbol{B} + M \right), \quad (5)$$

where $\boldsymbol{B}$ assigns positive values only to salient tokens at specific attention heads and $M$ is the attention mask. In VLMs,

these salient tokens typically correspond to visual tokens representing important image regions. Prior approaches either apply constant biases (Zhu et al., 2025a) or scale them proportionally to attention scores (Yin et al., 2025) at each decoding step.

In our work, for the selected layers $\mathcal{L}$, we enhance visual token attention for the top heads $\mathcal{H}_{OV}^l$ at each decoding step using the pre-computed saliency map. Unlike Yin et al. (2025), which relies on attention scores at the current step, our map is derived from user query processing, providing a holistic view of visual saliency with full query context. The formulation is defined as:

$$\hat{\boldsymbol{A}}_{h,-1,j}^l = \boldsymbol{A}_{h,-1,j}^l \cdot \exp(\alpha \hat{\mathcal{S}}_j), \quad (6)$$

where $l \in \mathcal{L}$, $h \in \mathcal{H}_{OV}^l$, $j \in X_V$, $\hat{\mathcal{S}}_j$ is the saliency score for visual token $j$, $\alpha$ is a scaling factor, and $-1$ denotes the current decoding position. After sum normalization, this is equivalent to Eq. 5 with bias term $\boldsymbol{B} = \alpha \hat{\mathcal{S}}_j$. To reduce the impact of outliers, we clip the saliency map $\hat{\mathcal{S}}$ in Eq. 3 at three standard deviations before min-max normalization, preventing overemphasis on any single region.

**Balancing Cross-modal Fusion.** Previous attention steering approaches (Yin et al., 2025; Zhu et al., 2025a) focus solely on visual tokens, neglecting the contribution of query tokens in cross-modal fusion. As shown in Figure 4, the attention proportions of query and visual tokens exhibit broadly aligned layer-wise trends, and both remain low even in layers with relatively higher proportions. Boosting only visual attention may improve grounding, but it risks weakening query comprehension, which is crucial for properly interpreting and utilizing visual information.

To preserve cross-modal fusion balance, we also scale query token attention proportionally to the overall visual attention enhancement. Formally:

$$\hat{\boldsymbol{A}}_{h,-1,j}^l = \boldsymbol{A}_{h,-1,j}^l \cdot \beta r^l, \quad l \in \mathcal{L}, \ h \in \mathcal{H}_{OT}^l, \ j \in X_T$$
$$r^l = \sum_{h \in \mathcal{H}_{OV}^l} \sum_{j \in X_V} \frac{\hat{\boldsymbol{A}}_{h,-1,j}^l}{\boldsymbol{A}_{h,-1,j}^l},$$
$$(7)$$

where $r^l$ quantifies the overall relative increase in visual attention at layer $l$, and $\beta$ is a scaling coefficient. After scaling, we normalize the enhanced attention matrix so that for each head and position, attention across all tokens sums to one: $\hat{\boldsymbol{A}}_{h,i,:}^l \leftarrow \hat{\boldsymbol{A}}_{h,i,:}^l / \sum_j \hat{\boldsymbol{A}}_{h,i,j}^l$.

# 5. Experiments

## 5.1. Experimental Setup

**Models and Baselines.** We evaluate our method on four models of varying architectures and sizes: LLaVA-1.5 7B

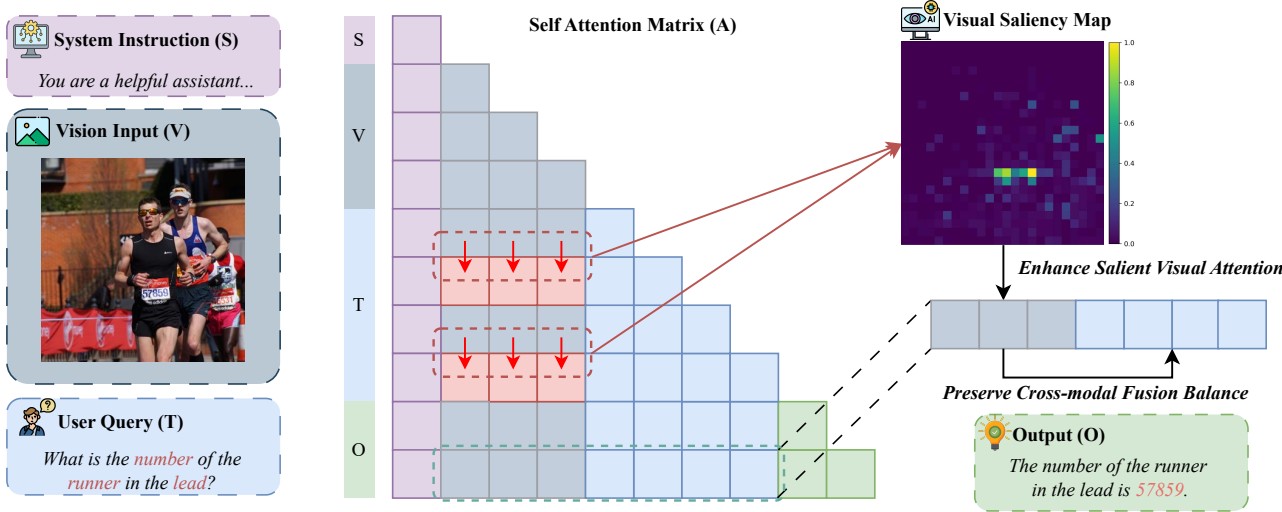

Figure 3. **Overview of GIFT.** GIFT computes a visual saliency map by tracking positive visual attention shifts ("gaze shifts") across information-rich query tokens during prefilling. During decoding, this map guides enhancement of salient visual attention while proportionally scaling query token attention to preserve cross-modal fusion balance.

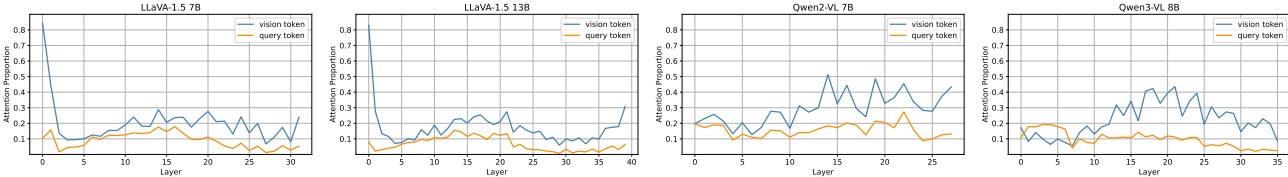

Figure 4. **Attention proportions to query and visual tokens from output tokens across layers.** Query and visual attention proportions follow similar patterns across layers, suggesting that effective cross-modal fusion relies on contributions from both modalities.

and 13B (Liu et al., 2023b), Qwen2-VL 7B (Wang et al., 2024b), and Qwen3-VL 8B (Bai et al., 2025). We compare against standard greedy decoding and four related methods. VAF (Yin et al., 2025) amplifies visual contribution by scaling visual token attention proportionally to their attention scores at each decoding step. VAR (Kang et al., 2025) identifies visual sink tokens based on model-specific hidden state dimensions and redistributes their attention proportionally to non-sink tokens in "image-centric" attention heads. MLLMs_know (Zhang et al., 2025) crops salient visual regions as additional inputs, using different strategies to identify salient regions; we evaluate its best-performing variant, Rel-Attn. VCD (Leng et al., 2024) contrasts output distributions of two input variants, one with original visual inputs and one with perturbed visual inputs, to reduce over-reliance on knowledge priors. Since these approaches do not provide Qwen2-VL or Qwen3-VL implementations or configurations, we compare our method with them only on the LLaVA-1.5 7B and 13B models.

**Benchmark and Metrics.** We evaluate on both vision-hallucination datasets and general vision-language benchmarks to assess effectiveness in reducing hallucination while maintaining reasoning capabilities, as overemphasizing vi-

sual perception can potentially impair reasoning capabilities. Vision-hallucination datasets cover three tasks: POPE (Li et al., 2023d) for object detection, evaluated using F1 and accuracy; CHAIR (Rohrbach et al., 2018) for image captioning, evaluated using CHAIRs and CHAIRi; and MMHal-Bench (Sun et al., 2023) for vision question answering, evaluated using hallucination rate and informativeness score. General vision-language benchmarks include MME (Fu et al., 2023) and SEED-Bench (Li et al., 2023a), both evaluated using accuracy. Further details are in Appendix B.

**Implementation Details.** Visual saliency maps are computed at the peak layer identified in Figure 2: 11 for LLaVA-1.5 7B, 10 for LLaVA-1.5 13B, 14 for Qwen2-VL 7B, and 18 for Qwen3-VL 8B. We tune enhancement coefficient $\alpha$ and enhancement layers $\mathcal{L}$ using the method in Appendix D across models. We set $\alpha$ to 5.0 for LLaVA models and 4.0 for Qwen models. The higher value for LLaVA reflects its lower original visual and query token attention, as shown in Figure 4. Enhancement layers are 12-22, 14-20, 5-18, and 13-23, respectively. We set $\beta$ to 1.0 for all models to preserve the original cross-modal balance. Hyperparameter robustness is studied in Section 5.3. Additional implementation details are in Appendix C.

*Table 2.* **Performance on vision-hallucination benchmarks.** GIFT outperforms all baselines on LLaVA models where direct comparisons are available, and consistently improves over greedy decoding across all models including Qwen. Best results are highlighted in bold.

| Model | Method | CHAIR | | POPE | | MMHal-Bench | |
|---|---|---|---|---|---|---|---|
| | | $C_s$ ($\downarrow$) | $C_i$ ($\downarrow$) | F1 ($\uparrow$) | Acc. ($\uparrow$) | Hal. ($\downarrow$) | Score ($\uparrow$) |
| LLaVA-1.5 7B | Greedy | 50.2 | 15.4 | 82.4 | 79.5 | 65.2 | 2.22 |
| | VAF | 49.6 | 14.3 | 81.0 | 77.2 | 66.3 | 2.16 |
| | Rel-Attn | 49.0 | 13.6 | 82.0 | 78.3 | 63.7 | 2.19 |
| | VAR | 54.0 | 15.5 | 83.1 | 80.1 | 60.8 | 2.40 |
| | VCD | 52.2 | 16.3 | 80.9 | 77.7 | 60.5 | 2.37 |
| | **GIFT** | **39.8** | **10.6** | **83.8** | **81.9** | **57.3** | **2.48** |
| LLaVA-1.5 13B | Greedy | 46.8 | 13.1 | 81.7 | 78.2 | 56.2 | 2.61 |
| | VAF | 47.4 | 13.2 | 80.6 | 76.4 | 59.2 | 2.46 |
| | Rel-Attn | 44.6 | 13.2 | 81.5 | 77.8 | 65.6 | 2.15 |
| | VAR | 51.8 | 14.0 | **82.2** | 78.5 | 56.2 | 2.52 |
| | VCD | 50.8 | 15.0 | 80.6 | 76.9 | 61.5 | 2.31 |
| | **GIFT** | **39.6** | **11.9** | 82.1 | **78.9** | **55.8** | **2.72** |
| Qwen2-VL 7B | Greedy | 24.8 | 9.1 | 86.0 | 86.5 | 32.7 | 3.53 |
| | **GIFT** | **21.2** | **7.7** | **86.8** | **86.9** | **27.5** | **3.58** |
| Qwen3-VL 8B | Greedy | 51.4 | 10.6 | 88.9 | 88.5 | 28.3 | 4.80 |
| | **GIFT** | **49.4** | **9.3** | **89.1** | **88.7** | **26.4** | **4.84** |

## 5.2. Experimental Results

**Hallucination Mitigation Performance.** Table 2 presents performance on vision-hallucination datasets. GIFT consistently outperforms all available baselines across datasets and models of varying architectures and sizes. Compared to greedy decoding, GIFT achieves improvements of up to 20.7% on CHAIR, 15.9% on MMHal-Bench, and 3.0% on POPE, while also improving MMHal-Bench informativeness score by 11.7%. These results demonstrate its effectiveness and robustness in mitigating hallucinations in vision-language models across diverse evaluation settings. Qualitative examples from MMHal-Bench are provided in Appendix I.

We observe that Qwen3-VL 8B, despite achieving the best performance on POPE and MMHal-Bench, shows higher CHAIR scores than other models. We attribute this to its tendency to generate detailed, reasoning-based descriptions that may be penalized by CHAIR's string-matching approach against predefined objects, suggesting the metric may need refinement for modern models producing nuanced outputs.

**General Vision-Language Performance.** As shown in Table 3, GIFT consistently maintains performance comparable to greedy decoding on SEED-Bench and MME benchmarks across all models, demonstrating minimal impact on general capabilities. Since these benchmarks primarily evaluate reasoning and general knowledge rather than visual perception accuracy, we do not expect GIFT to improve per-

*Table 3.* **Performance on general vision-language benchmarks.** GIFT maintains performance comparable to greedy decoding, while baselines show mixed results.

| Method | LLaVA-1.5 7B | | LLaVA-1.5 13B | | Qwen2-VL 7B | | Qwen3-VL 8B | |
|---|---|---|---|---|---|---|---|---|
| | MME | SEED | MME | SEED | MME | SEED | MME | SEED |
| Greedy | 1751.6 | 65.5 | 1807.5 | 67.8 | 2278.9 | 76.0 | 2403.5 | 78.4 |
| VAF | 1787.6 | 64.9 | 1815.4 | 67.0 | - | - | - | - |
| Rel-Attn | 1811.3 | 65.0 | 1758.5 | 66.7 | - | - | - | - |
| VAR | 1780.8 | 65.4 | 1782.5 | 67.9 | - | - | - | - |
| VCD | 1742.3 | 63.5 | 1804.0 | 66.3 | - | - | - | - |
| **GIFT** | 1750.5 | 65.6 | 1815.9 | 67.7 | 2279.1 | 76.0 | 2418.5 | 78.3 |

formance. In contrast, baseline methods show mixed results, with some exhibiting performance degradation, highlighting the challenge of mitigating hallucination while preserving general performance.

**Latency Analysis.** We benchmark GIFT's computational overhead using LLaVA-1.5 7B on MMHal-Bench, measuring latency relative to greedy decoding with a fixed output length of 32 tokens, which approximates the average output length under greedy decoding. As shown in Table 4, GIFT runs at only 1.13× the latency of greedy decoding, compared to 1.56× for Rel-Attn, 1.99× for VCD, and 11.10× for VAR. While VAF is slightly faster at 1.01×, GIFT consistently outperforms all baselines across vision-hallucination benchmarks and models, offering a strong tradeoff between efficiency and performance. We list potential latency optimization approaches as future work in Appendix G.

*Table 4.* **Inference latency comparison.** Latency relative to greedy decoding (1.00×) for generating 32 tokens. GIFT incurs low computational overhead (1.13×).

| Method | **GIFT** | VAF | Rel-Attn | VCD | VAR |
|---|---|---|---|---|---|
| Relative Latency | 1.13× | 1.01× | 1.56× | 1.99× | 11.10× |

*Table 5.* **Ablation study on enhancement strategies.** GIFT, which increases both visual and query attention, outperforms variants that only increase (Inc. V.) or only calibrate (Cal. V.) visual attention.

| Model | Setup | MMHal-Bench | | POPE | |
|---|---|---|---|---|---|
| | | Hal. (↓) | Score (↑) | F1 (↑) | Acc. (↑) |
| LLaVA-1.5 7B | Inc. V. | 60.8 | 2.36 | 82.3 | 79.3 |
| | Cal. V. | 61.5 | 2.32 | 82.4 | 79.5 |
| | **GIFT** | **57.3** | **2.48** | **83.8** | **81.9** |
| LLaVA-1.5 13B | Inc. V. | 59.2 | 2.51 | 81.3 | 77.6 |
| | Cal. V. | 59.8 | 2.46 | 81.6 | 78.0 |
| | **GIFT** | **55.8** | **2.72** | **82.1** | **78.9** |
| Qwen2-VL 7B | Inc. V. | 35.2 | 3.41 | 85.3 | 86.0 |
| | Cal. V. | 31.9 | 3.56 | 85.8 | 86.4 |
| | **GIFT** | **27.5** | **3.58** | **86.8** | **86.9** |
| Qwen3-VL 8B | Inc. V. | 35.4 | 4.50 | 88.7 | 88.1 |
| | Cal. V. | 28.2 | 4.77 | 88.9 | 88.5 |
| | **GIFT** | **26.4** | **4.84** | **89.1** | **88.7** |

## 5.3. Ablations

We ablate four aspects of GIFT: the joint contribution of visual attention enhancement and cross-modal fusion balance, the impact of the enhancement strength $\alpha$, the choice of enhancement layers, and the choice of information-rich word extraction strategy.

**Contribution of Visual Enhancement and Fusion Balance.** We perform ablation studies to evaluate the joint contribution of two components: (1) enhancing visual token attention in proportion to task-relevant saliency, and (2) preserving cross-modal fusion balance. We consider two variants: one that increases visual attention without maintaining fusion balance by omitting Eq.7 (Inc. V), and another that calibrates visual attention distribution to emphasize salient tokens while keeping the overall visual contribution unchanged (Cal. V.). In the latter, the enhanced visual attention $\hat{\mathcal{A}}^l$ is scaled down by $r^l$ from Eq. 7, leaving query token attention unchanged. As shown in Table 5, our full method, GIFT, consistently outperforms both variants on POPE and MMHal-Bench benchmarks across all models by up to 25.4%, demonstrating that both components are essential for effective hallucination mitigation.

**Impact of Enhancement Strength.** We evaluate GIFT with enhancement coefficient $\alpha$ ranging from 1.0 to 7.0 on POPE and MME benchmarks, with a step size of 1.0. Figure 5 shows results for LLaVA-1.5 7B. As $\alpha$ increases, hallucination mitigation steadily improves on POPE while

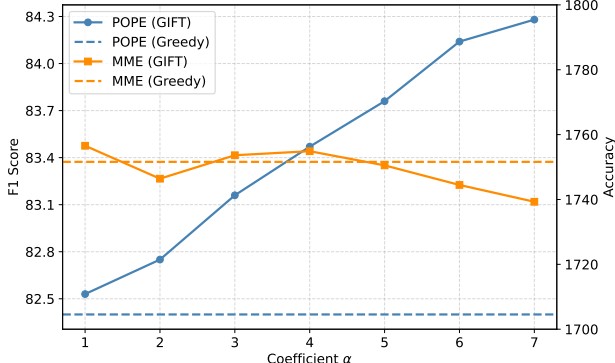

*Figure 5.* **Impact of enhancement coefficient $\alpha$ on LLaVA-1.5 7B.** Higher values improve hallucination mitigation (POPE) but degrade reasoning (MME) when overly large.

reasoning on MME stays close to greedy decoding across a broad range of moderate values. At excessively large $\alpha$, however, POPE plateaus and MME drops below greedy decoding, indicating that over-amplifying visual attention sacrifices reasoning for perceptual detail. Similar trends are observed across models, detailed in Appendix D.

*Table 6.* **Ablation on enhancement layer ranges for LLaVA-1.5 7B.** Multiple moderate configurations (12-22 or narrower) achieve strong performance, demonstrating robustness to layer selection.

| Layer Range | None | 16-18 | 14-20 | **12-22** | 10-24 | 8-26 | 6-28 | 4-30 |
|---|---|---|---|---|---|---|---|---|
| POPE | 82.4 | 83.0 | 83.2 | **83.8** | 83.6 | 83.9 | 83.8 | 84.0 |
| SEED | 65.5 | 65.7 | 65.6 | **65.6** | 65.2 | 65.1 | 64.6 | 63.9 |

**Sensitivity to Enhancement Layer Selection.** We progressively expand or contract the enhancement layer range from the default by adjusting each boundary by 2 layers. Table 6 shows results for LLaVA-1.5 7B, where the default range is 12-22. All configurations outperform the baseline on POPE, with performance increasing as ranges widen due to stronger perceptual focus. However, excessively wide ranges (10-24 and beyond) degrade reasoning on SEED-Bench, suggesting over-prioritization of perceptual grounding. Notably, multiple moderate configurations (12-22 and narrower) achieve strong hallucination mitigation while preserving reasoning performance, demonstrating robustness to enhancement layer selection.

**Choice of Information-Rich Word Extraction.** We compare our default POS tagging-based extraction with an LLM-based alternative that prompts Qwen3-VL 8B to identify query tokens requiring visual grounding (full prompt in Appendix F). We evaluate both strategies on MMHal-Bench, chosen for its diverse question types. As shown in Table 7, the LLM-based approach yields stronger hallucination mitigation across both LLaVA-1.5 7B and Qwen2-VL 7B, suggesting that it selects information-rich tokens more accu-

rately than POS tagging. We adopt POS tagging by default for its negligible computational overhead, but recommend the LLM-based variant when stronger hallucination mitigation justifies the additional cost

*Table 7.* **Comparison of information-rich word extraction strategies on MMHal-Bench.** The LLM-based approach achieves stronger hallucination mitigation than POS tagging at additional cost.

| Model | Method | Hal. ($\downarrow$) | Score ($\uparrow$) |
|---|---|---|---|
| LLaVA-1.5 7B | POS Tag | 57.3 | 2.48 |
| | LLM-Based | **52.6** | **2.60** |
| Qwen2-VL 7B | POS Tag | 27.5 | 3.58 |
| | LLM-Based | **26.0** | **3.67** |

## 6. Conclusion

We identify critical limitations that existing inference-time hallucination mitigation methods for vision-language models (VLMs) fail to address simultaneously, including visual attention sink, low visual contribution, and imbalanced cross-modal fusion. To address these challenges, we introduce **G**aze Sh**i**ft-Guided Cross-modal **F**usion Enhancemen**t** (**GIFT**), which constructs a holistic visual saliency map by tracking "gaze shifts" during user query processing and uses it to enhance both visual and query attentions at each decoding step. Extensive experiments demonstrate that GIFT reduces hallucination by up to 20.7% across models and datasets, while maintaining general vision-language reasoning performance with low computational overhead.

A primary limitation is GIFT's reliance on the query to identify relevant visual regions. When answering does not depend on visual information, or when substantial query portions are unrelated to the image, the method may produce inaccurate saliency maps, causing incorrect focus. Future work could enhance vision-relevant token identification through LLM prompting or a lightweight classifier, where token scores weight their contribution to saliency computation. When no tokens relate to the image, saliency computation could be bypassed entirely.

## Impact Statement

This paper presents work whose goal is to advance the field of machine learning and foundation models by reducing hallucinations in vision-language models (VLMs). Hallucination, i.e., content that cannot be substantiated by either textual or visual inputs, pose significant risks in high-stakes domains such as biomedicine, autonomous driving, and robotics. By mitigating hallucination in VLM outputs, our work has the potential to make systems in these domains safer and more trustworthy for real-world deployment. How-

ever, we acknowledge that our work does not eliminate all risks associated with VLM development and deployment. Our research complies with all legal and ethical standards, and we have not identified any negative effects that necessitate highlighting in this discussion.

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

# A. Robustness of Extraction Layer Selection

We assess the robustness of our saliency map extraction layer selection along two axes: (i) different random samples drawn from the same dataset, and (ii) different datasets. We consider TextVQA (Singh et al., 2019) (visual question answering) and MathVista (Lu et al., 2024) (math reasoning in visual contexts), which contrast substantially in task type, and run the procedure on LLaVA-1.5 7B and Qwen3-VL 8B. For each (model, dataset) combination, we repeat the analysis from Section 3 with five random seeds, each sampling 50 examples, and compute the sum of $\hat{S}^l$ without min-max normalization. Tables 8–10 report values for layers surrounding the peak.

**Stability across samples.**   Within each (model, dataset) combination, the peak layer is identical in all five runs: layer 11 for LLaVA-1.5 7B and layer 18 for Qwen3-VL 8B. The peak layer is therefore not sensitive to which examples are sampled.

**Stability across datasets.**   For each model, the peak layer on TextVQA matches the peak layer on MathVista, again layer 11 for LLaVA-1.5 7B and layer 18 for Qwen3-VL 8B. The relative ordering of neighboring layers is also preserved across datasets. Combined with sample-level stability, this supports our claim that the peak layer is an intrinsic property of each model rather than dataset-specific behavior.

| Layer | 6 | 7 | 8 | 9 | 10 | 11 | 12 | 13 | 14 | 15 | 16 |
|---|---|---|---|---|---|---|---|---|---|---|---|
| Run 1 | 0.100 | 0.102 | 0.162 | 0.143 | 0.183 | **0.215** | 0.153 | 0.116 | 0.162 | 0.099 | 0.098 |
| Run 2 | 0.100 | 0.102 | 0.160 | 0.139 | 0.184 | **0.208** | 0.149 | 0.112 | 0.163 | 0.103 | 0.096 |
| Run 3 | 0.098 | 0.104 | 0.160 | 0.143 | 0.184 | **0.213** | 0.150 | 0.116 | 0.165 | 0.102 | 0.101 |
| Run 4 | 0.097 | 0.103 | 0.158 | 0.141 | 0.183 | **0.211** | 0.148 | 0.116 | 0.164 | 0.102 | 0.100 |
| Run 5 | 0.093 | 0.095 | 0.152 | 0.137 | 0.186 | **0.210** | 0.155 | 0.117 | 0.162 | 0.103 | 0.100 |

*Table 8.* LLaVA-1.5 7B on TextVQA. Sum of visual attention shifts $\hat{S}^l$ for layers surrounding the peak across five runs. Layer 11 (bold) consistently exhibits the highest value across all runs.

| Layer | 6 | 7 | 8 | 9 | 10 | 11 | 12 | 13 | 14 | 15 | 16 |
|---|---|---|---|---|---|---|---|---|---|---|---|
| Run 1 | 0.042 | 0.048 | 0.073 | 0.065 | 0.073 | **0.101** | 0.073 | 0.059 | 0.080 | 0.055 | 0.060 |
| Run 2 | 0.041 | 0.049 | 0.075 | 0.065 | 0.074 | **0.101** | 0.072 | 0.060 | 0.079 | 0.054 | 0.059 |
| Run 3 | 0.039 | 0.046 | 0.069 | 0.061 | 0.070 | **0.092** | 0.065 | 0.055 | 0.075 | 0.052 | 0.054 |
| Run 4 | 0.041 | 0.047 | 0.070 | 0.063 | 0.071 | **0.095** | 0.068 | 0.057 | 0.076 | 0.054 | 0.055 |
| Run 5 | 0.051 | 0.059 | 0.080 | 0.072 | 0.100 | **0.122** | 0.082 | 0.062 | 0.086 | 0.063 | 0.065 |

*Table 9.* LLaVA-1.5 7B on MathVista. Sum of visual attention shifts $\hat{S}^l$ for layers surrounding the peak across five runs. Layer 11 (bold) consistently exhibits the highest value across all runs.

| Layer | 13 | 14 | 15 | 16 | 17 | 18 | 19 | 20 | 21 | 22 | 23 |
|---|---|---|---|---|---|---|---|---|---|---|---|
| Run 1 | 0.372 | 0.237 | 0.318 | 0.227 | 0.309 | **0.404** | 0.276 | 0.281 | 0.287 | 0.130 | 0.165 |
| Run 2 | 0.372 | 0.242 | 0.341 | 0.232 | 0.315 | **0.412** | 0.286 | 0.289 | 0.290 | 0.125 | 0.169 |
| Run 3 | 0.379 | 0.254 | 0.347 | 0.245 | 0.323 | **0.427** | 0.296 | 0.290 | 0.297 | 0.128 | 0.159 |
| Run 4 | 0.379 | 0.250 | 0.352 | 0.243 | 0.312 | **0.414** | 0.296 | 0.298 | 0.307 | 0.132 | 0.169 |
| Run 5 | 0.364 | 0.247 | 0.331 | 0.246 | 0.318 | **0.404** | 0.290 | 0.282 | 0.292 | 0.132 | 0.170 |

*Table 10.* Qwen3-VL 8B on MathVista. Sum of visual attention shifts $\hat{S}^l$ for layers surrounding the peak across five runs. Layer 18 (bold) consistently exhibits the highest value across all runs.

# B. Datasets

**POPE.**   The Polling-based Object Probing Evaluation (POPE) (Li et al., 2023d) assesses object hallucination in VLMs using binary questions (e.g., "Is there a frisbee in the image?"). Objects are drawn from three splits: *random*, *popular*, and *adversarial*, corresponding respectively to randomly chosen missing objects, frequently occurring objects, and co-occurring but absent objects. POPE uses images from MSCOCO (Lin et al., 2014), A-OKVQA (Schwenk et al., 2022), and GQA (Hudson & Manning, 2019), resulting in nine splits, each containing 500 MSCOCO images with six questions per image. Results are reported as macro-averaged F1 and accuracy over all splits.

| Layer | 13 | 14 | 15 | 16 | 17 | 18 | 19 | 20 | 21 | 22 | 23 |
|-------|-----|-----|-----|-----|-----|-----|-----|-----|-----|-----|-----|
| Run 1 | 0.118 | 0.073 | 0.105 | 0.080 | 0.102 | **0.124** | 0.095 | 0.087 | 0.098 | 0.054 | 0.063 |
| Run 2 | 0.130 | 0.080 | 0.116 | 0.088 | 0.113 | **0.135** | 0.106 | 0.101 | 0.110 | 0.060 | 0.071 |
| Run 3 | 0.126 | 0.079 | 0.113 | 0.088 | 0.108 | **0.133** | 0.099 | 0.093 | 0.103 | 0.057 | 0.067 |
| Run 4 | 0.126 | 0.082 | 0.114 | 0.089 | 0.113 | **0.138** | 0.105 | 0.100 | 0.109 | 0.063 | 0.073 |
| Run 5 | 0.125 | 0.080 | 0.114 | 0.087 | 0.112 | **0.135** | 0.102 | 0.096 | 0.105 | 0.059 | 0.068 |

*Table 11.* Qwen3-VL 8B on TextVQA. Sum of visual attention shifts $\hat{\mathcal{S}}^l$ for layers surrounding the peak across five runs. Layer 18 (bold) consistently exhibits the highest value across all runs.

**CHAIR.** CHAIR (Captioning Hallucination Assessment with Image Relevance) (Rohrbach et al., 2018) contains 500 images for evaluating hallucination in image captioning. We use two metrics: $C_I$, which measures the proportion of hallucinated objects among all mentioned objects in captions, and $C_S$, which measures the proportion of captions containing at least one hallucinated object. Formally, these are defined as:

$$C_I = \frac{|\text{hallucinated objects}|}{|\text{all mentioned objects}|}, \quad C_S = \frac{|\text{captions with hallucinated objects}|}{|\text{all captions}|}$$

**MMHal-Bench.** MMHal-Bench (Sun et al., 2023) is a benchmark designed to evaluate hallucination in vision language models (VLMs). It contains 96 challenging questions based on images from the OpenImages dataset (Kuznetsova et al., 2020), each paired with a corresponding ground-truth answers and annotated image content. Model responses are scored using GPT-4 through a pre-defined prompt that assesses both informativeness and hallucination.

**MME.** MME (Fu et al., 2023) is a benchmark designed to assess both perception and cognition capabilities of vision language models across 14 subtasks. Each subtask evaluates a specific aspect of visual understanding or reasoning capability. For all experiments, We report performance using the accuracy metric as defined in the original paper.

**SEED-Bench.** SEED-Bench (Li et al., 2023a) is a comprehensive benchmark designed to evaluates general vision-language reasoning capabilities. It contains 19,000 multiple-choice questions spanning 12 evaluation dimensions, covering both image and video modalities. In this work, we focus exclusively on the image modality and report model performance using accuracy.

## C. Implementation Details

For all experiments, we employ greedy decoding with eager attention computation, and run inference on a single NVIDIA A100 Tensor Core GPU (40GB) instance to ensure reproducibility and fair comparisons across models and baselines. We use float16 precision for LLaVA models and bfloat16 for Qwen models.

For POPE, different baselines append varying suffixes to the questions, such as "Please just answer yes or no." or "Answer the question using a single word or phrase.", leading to substantial variation in evaluation results. To ensure fair comparison, we use the original dataset questions without modification. For MMHal-Bench, since the original GPT-4 version has been deprecated, we use GPT-4.1 (*gpt-4.1-2025-04-14*) for scoring. To account for the inherent randomness in GPT-4.1 scoring outputs, each evaluation is repeated five times, and the results are averaged. We set the max number of new tokens to 10 for POPE, MME, and SEED-Bench, and to 1024 for CHAIR and MMHal-Bench.

## D. Hyperparameter Tuning

Our method involves four key hyperparameters: (1) the layer used to compute the visual saliency map, (2) the layers selected for cross-modal fusion enhancement, (3) the visual attention enhancement coefficient $\alpha$, and (4) the query attention enhancement coefficient $\beta$.

**Layer for Computing Visual Saliency Map.** We select the saliency map computation layer as the one exhibiting the largest positive changes in visual token attention when processing information-rich user query tokens, as discussed in Section 3. Based on Figure 2, this corresponds to layer 11 for LLaVA-1.5 7B, layer 10 for LLaVA-1.5 13B, layer 14 for

*Figure 6.* Performance on the POPE and MME datasets with varying enhancement coefficients $\alpha$ across models.

Qwen2-VL 7B, and layer 18 for Qwen3-VL 8B. We conduct experiments in Appendix A to demonstrate that this layer remains stable across different data samples.

**Layers for Cross-Modal Fusion Enhancement.** Since the benchmarks we consider lack dedicated validation sets for hyperparameter tuning, we follow Kang et al. (2025) by randomly sample 10% of the POPE and MME datasets as "pseudo-validation" sets for tuning, applying the resulting hyperparameters to all benchmark samples. Given the large number of transformer layers across models, we restrict the grid search for cross-modal fusion enhancement layers: the start layer is chosen between the first layer reaching a visual attention proportion of 0.2 and the peak layer, and the end layer is chosen between the peak layer and the last layer reaching 0.2, based on Figure 4. Following this procedure, we select layers 12-22 for LLaVA-1.5 7B, layers 14-20 for LLaVA-1.5 13B, layers 5-18 for Qwen2-VL 7B, and layers 13-23 for Qwen3-VL 8B. We conduct ablation studies in Section 5.3 to demonstrate that our method is robust to enhancement layer selection.

**Visual Attention Enhancement Coefficient.** To tune the visual attention enhancement coefficient $\alpha$, we vary its value from 1.0 to 7.0 on the POPE and MME datasets, evaluating the trade-off between hallucination mitigation and reasoning performance. Results across all four models are shown in Figure 6. Based on these results, we set $\alpha = 5.0$ for LLaVA models, and $\alpha = 4.0$ for Qwen models. The higher value required for LLaVA reflects its lower original visual and query token attention, as shown in Figure 4, which necessitates stronger enhancement.

**Query Attention Enhancement Coefficient.** For query token attention enhancement, we set $\beta = 1.0$ to preserve the original cross-modal fusion balance. Investigating whether this balance is truly optimal is left for future work.

# E. More Experiments on GIFT

To complement the main results, we evaluate GIFT on two additional benchmarks using the same hyperparameters as in the main experiments, without any per-benchmark tuning: AMBER (Wang et al., 2023a) for hallucination and MMBench (Liu et al., 2024d) for general vision-language reasoning.

*Table 12.* **Performance on AMBER.** GIFT outperforms greedy decoding across models. Best results are highlighted in bold.

| Model | Method | AMBER ($\uparrow$) | Discriminative ($\uparrow$) | Generative ($\downarrow$) |
|---|---|---|---|---|
| LLaVA-1.5 7B | Greedy | 83.9 | 74.9 | 7.2 |
| | GIFT | **87.2** | **80.9** | **6.6** |
| LLaVA-1.5 13B | Greedy | 83.4 | 73.5 | 6.7 |
| | GIFT | **85.7** | **76.8** | **5.5** |
| Qwen2-VL 7B | Greedy | 84.1 | 74.4 | 6.2 |
| | GIFT | **85.4** | **76.2** | **5.5** |
| Qwen3-VL 8B | Greedy | 80.2 | 68.3 | 7.9 |
| | GIFT | **83.6** | **74.0** | **6.8** |

**AMBER.** AMBER (Wang et al., 2023a) is a hallucination benchmark covering existence, attribute, and relation hallucinations through two task formats: a discriminative task with yes/no questions over each hallucination type and a generative task that asks the model to describe an image and scores hallucinations against ground-truth annotations. We report the discriminative and generative subscores together with the aggregate AMBER score that combines them. As shown in

Table 12, GIFT outperforms greedy decoding on every model and metric, with up to a 4.2-point gain on the aggregate score, extending the hallucination reductions seen on POPE, CHAIR, and MMHal-Bench.

**MMBench.**    MMBench (Liu et al., 2024d) is a multiple-choice benchmark that evaluates vision-language models across a fine-grained taxonomy of perception and reasoning abilities, using a CircularEval strategy that rotates the position of the correct option to reduce sensitivity to choice ordering. As shown in Table 13, GIFT matches or slightly improves over greedy decoding on every model, consistent with the MME and SEED-Bench results in the main paper.

*Table 13.* **Performance on MMBench.** GIFT performs on par with greedy decoding across models.

| Model | Greedy | GIFT |
|---|---|---|
| LLaVA-1.5 7B | 73.1 | 73.1 |
| LLaVA-1.5 13B | 75.6 | 75.8 |
| Qwen2-VL 7B | 84.6 | 84.6 |
| Qwen3-VL 8B | 86.9 | 87.2 |

## F. Prompt for LLM-Based Information-Rich Word Extraction

For the LLM-based variant in the ablation in Section 5.3, we prompt Qwen3-VL 8B with the following instruction to identify query tokens that require visual grounding:

> *Identify all words (including objects, actions, visual attributes, and spatial relations) that require visual information from an image to answer this question. Return them in the order they appear in the question, comma-separated.*

The returned words are then mapped back to their corresponding token indices in the original query, and used as $X_{Tr}$ in Eq. 3 in place of the POS-tagged tokens.

## G. Limitations and Future Work

We acknowledge several limitations to address in future work.

First, our method relies heavily on the user query to identify relevant visual regions. When answering does not dependent on visual information, or when substantial portions of the query are unrelated to the image content (e.g., retrieved text documents), the method may produce inaccurate or unnecessary visual saliency maps, causing the model to focus on incorrect image regions or attend to visual information unnecessarily. To address this, we plan to enhance the identification of vision-relevant, information-rich query tokens through LLM prompting or by fine-tuning a lightweight auxiliary model for token classification, where each token's score weights its contribution to saliency map computation. When no query tokens relate to the image content, we will bypass visual saliency map computation entirely, preventing unnecessary computational overhead and potential error propagation. In addition, to our knowledge, no existing benchmark specifically targets scenarios where answering does not depend on visual information despite image input being provided, or where substantial query portions are unrelated to image content. Creating such a dataset would benefit the broader research community.

Second, while computational overhead is relatively low, inference is still 13% slower than standard greedy decoding. This overhead can be reduced by adopting more efficient attention computation mechanisms or pruning layers during visual saliency map computation, and by partially reusing the computed key-value caches for layers preceding the start of cross-modal fusion enhancement. These optimizations would make GIFT more practical for deployment while maintaining its effectiveness in reducing hallucinations.

Third, not all decoding steps require attention to visual inputs, as some steps primarily involve reasoning. Developing a method to dynamically determine when to enhance cross-modal fusion is left for future work.

## H. Additional Saliency Map Examples

Figure 7 shows two examples illustrating that GIFT's saliency map adapts to query intent even when the query does not explicitly name target objects. On the left, an open-ended CHAIR captioning sample of a man playing tennis: GIFT's

saliency map spans the person, the tennis ball, and the playfield, covering the key elements needed for a comprehensive description rather than fixating on a single object. On the right, an MMHal-Bench sample with the query *"What is the weather like in the image?"*, where the relevant visual evidence lies in the sky and surrounding background rather than any foreground object; GIFT's saliency map correctly concentrates on these background regions.

Q: *Please describe this image in detail.*

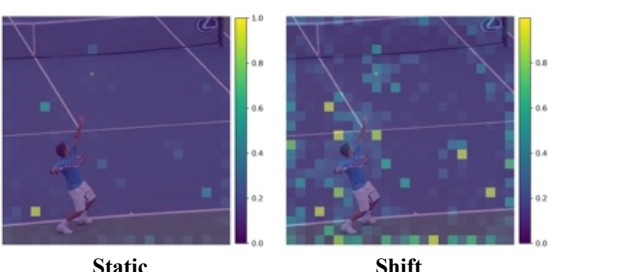

Q: *What is the weather like in the image?*

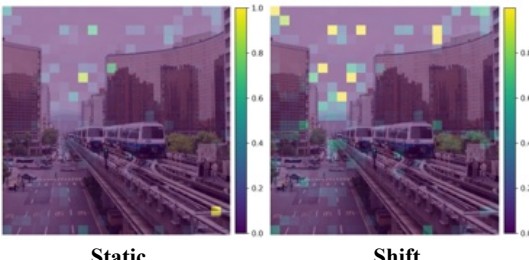

Static        Shift             Static        Shift

*Figure 7.* Additional GIFT saliency map examples. Left: a CHAIR captioning sample, where the saliency map spans the person, the tennis ball, and the playfield. Right: an MMHal-Bench sample with the query *"What is the weather like in the image?"*, where the saliency map focuses on the sky and background.

## I. Qualitative Analysis

Figure 8-12 illustrate example outputs from the MMHal-Bench dataset, comparing our method, GIFT, with standard greedy decoding. GIFT effectively mitigates hallucination by accurately describing facts present in the image and correctly identifying absent entities. For instance, in Figure 9, the user query asks for a description of the person playing tennis, while the image only contains a tennis racket and a ball without any person. GIFT correctly responds that no person is present, whereas greedy decoding assumes a person is in the scene and hallucinates details based on visual cues and model's knowledge priors. This improvement can be attributed to GIFT's cross-modal fusion balance, which enhances both visual and query attention for more accurate and tightly integrated representations of visual and textual information. Figure 12 shows that, although GIFT generates fewer hallucinations than greedy decoding, some errors remain, highlighting the need for further refinement.

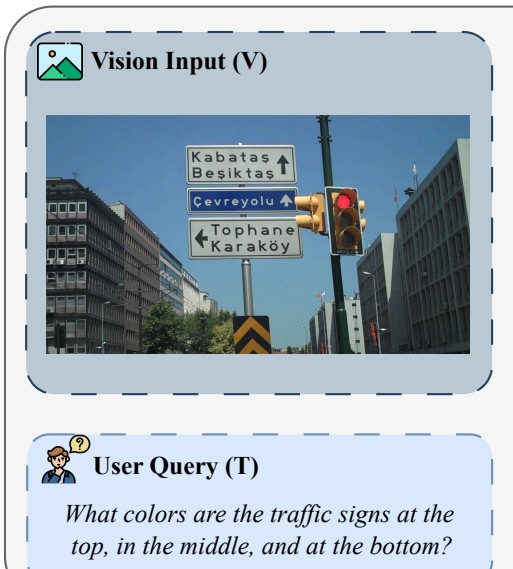

**Vision Input (V)**

**User Query (T)**

*What colors are the traffic signs at the top, in the middle, and at the bottom?*

**Ground Truth:**
The traffic signs at the top and bottom are white, while the one in the center is blue.

**Greedy Decoding:**
The traffic signs at the top, **in the middle**, and at the bottom are **white with black text**.

**GIFT (Ours):**
The top, the traffic sign is white with black text. In the middle, the traffic sign is blue with white text. At the bottom, the traffic sign is white with black text.

*Figure 8.* An example from the MMHal-Bench dataset. Hallucinations are highlighted in red.

**Vision Input (V)**

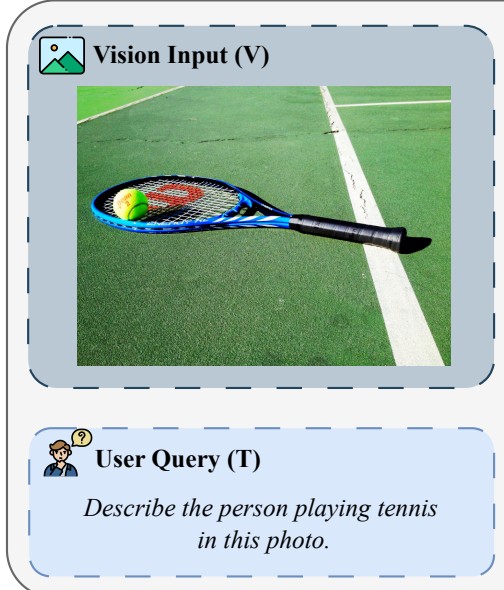

**User Query (T)**

*Describe the person playing tennis
in this photo.*

**Ground Truth:**
No, there is only a tennis racket and a ball on the ground. No one is playing tennis in this photo.

**Greedy Decoding:**
The the photo, there is a person playing tennis. The person is holding a blue and white tennis racket with a black grip. They are standing on a green tennis court, and the ball is in the air, indicating that they are in the middle of a serve or a return. The person is wearing a white shirt and black shorts, and they appear to be focused on the game.

**GIFT (Ours):**
There is no person in the photo.

*Figure 9.* An example from the MMHal-Bench dataset. Hallucinations are highlighted in red.

**Vision Input (V)**

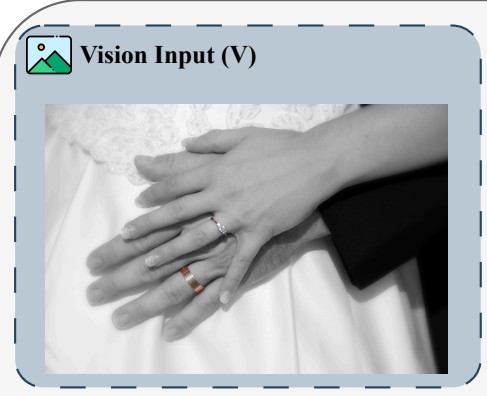

**User Query (T)**

*Describe the details in this photo.*

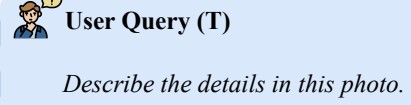

**Ground Truth:**
In the photo, a woman's hand is holding a man's hand. The woman's hand is wearing a diamond wedding ring, while the man's hand is wearing a simple wedding band. It seems that the woman is in a white dress and the man is in a black suit, but only a part can be seen in this photo. The image is black and white, except the wedding ring and band. This photo is likely taken in a wedding.

**Greedy Decoding:**
The image is a black-and-white photograph that captures a close-up view of two hands resting on a white surface, likely a wedding dress. The hands are positioned in a way that they overlap, **with the left hand on top of the right hand**. Both hands are adorned with rings, indicating a wedding or engagement setting. The left hand, which is on top, has a ring on the ring finger, **while the right hand, which is underneath, has a ring on the middle finger**. The background is blurred, focusing attention on the hands and the rings. The overall composition suggests a moment of celebration or commitment, likely a wedding ceremony.

**GIFT (Ours):**
The image is a black and white photograph that captures a close-up view of two hands resting on top of each other. The hands belong to a bride and groom, as indicated by the presence of wedding rings. The bride's hand is adorned with a diamond engagement ring and a wedding band, while the groom's hand features a simple wedding band. The background is a white fabric, likely the bride's wedding dress, which adds to the formal and celebratory atmosphere of the scene. The photograph focuses on the the hands and rings, emphasizing the significance of the wedding ceremony.

*Figure 10.* An example from the MMHal-Bench dataset. Hallucinations are highlighted in red.

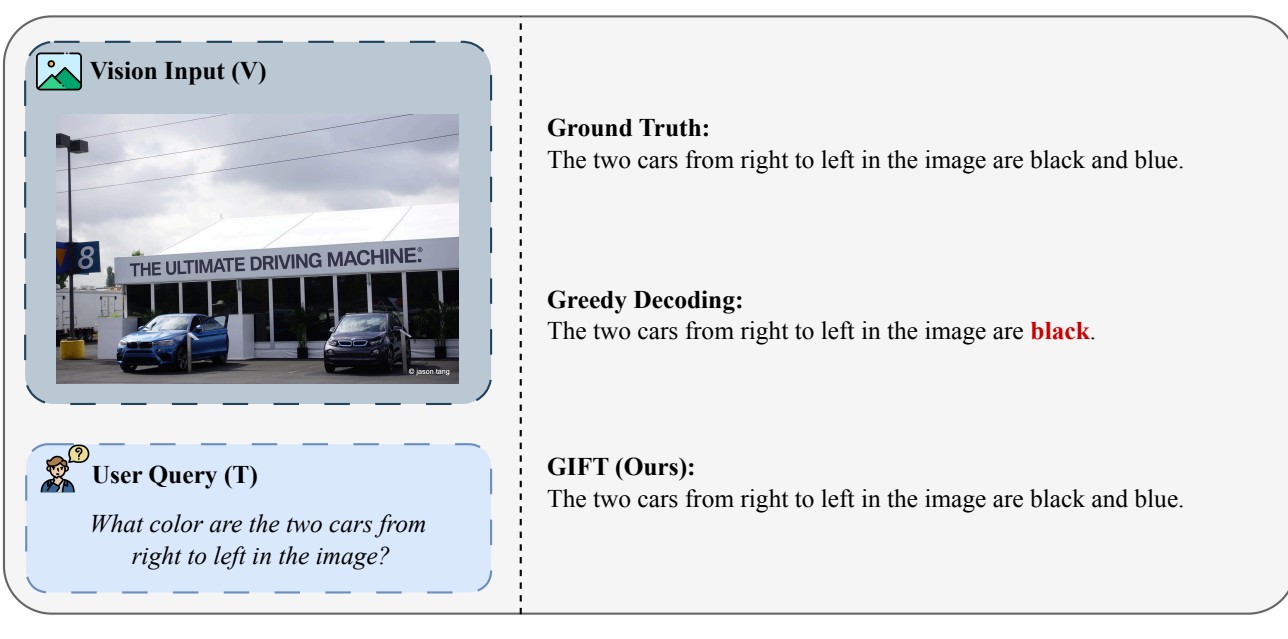

*Figure 11.* An example from the MMHal-Bench dataset. Hallucinations are highlighted in red.

**Vision Input (V)**

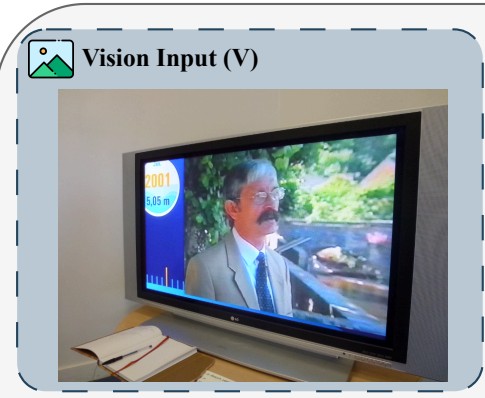

**User Query (T)**

*Describe the image in detail.*

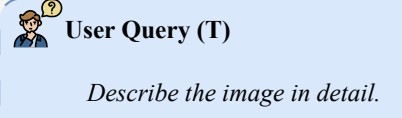

**Ground Truth:**
In this image, I see a book, a pen, and a television on a table. The TV screen shows a man wearing a tie and a suit speaking in a park.

**Greedy Decoding:**
The image shows a large flat-screen television **mounted on a wall**. The television is displaying a scene with a man in a suit and tie, who appears to be speaking or present. The man is standing in front of a backdrop that includes greenery and a body of water, possibly a river or a lake. The screen also shows a graphical representation of water levels, with a blue bar indicating the water level at 5.05 meters in **January 2**. The television is placed on a wooden surface, possibly a desk or a table, and **there is a notebook and a pen on the surface to the left of the television**. The notebook is open, and the pen is resting on it. The overall setting suggests a casual or professional environment, possibly an office or a study room.

**GIFT (Ours):**
The image depicts a television set **mounted on a wall**, displaying a scene with a man in a suit and tie. The man appears to be speaking or present in a formal setting, **possibly a newsroom or a professional environment**. The television screen shows a graphical representation of water levels, with a blue bar indicating the water level at 5.05 meters. The background of the screen includes greenery, suggesting an outdoor or natural setting. \n\nIn the foreground, there is a desk with a notebook and a pen on it. The notebook is open, and the pen is placed on top of it. The desk appears to be made of wood, and the notebook is white with black lines, indicating it is likely a standard lined notebook. The pen is black and appears to be a standard ballpoint pen. The overall setting suggests a workspace or a study area.

*Figure 12.* An example from the MMHal-Bench dataset. Hallucinations are highlighted in red.

