# OpenReview forum: "Capturing Gaze Shifts for Guidance: Cross-Modal Fusion Enhancement for VLM Hallucination Mitigation"
_ICML.cc/2026/Conference — ICML 2026 regular_

### Official Review · Reviewer_5Rxy · 2026-03-12

**Soundness:** 4
**Presentation:** 4
**Significance:** 3
**Originality:** 3
**Overall Recommendation:** 4
**Confidence:** 3

**Summary:**

This paper studies hallucination in vision-language models (VLMs) and proposes an inference-time mitigation method called Gaze Shift-Guided Cross-modal Fusion Enhancement (GIFT). The authors observe that existing attention-steering approaches either overemphasize visual tokens or ignore issues such as visual attention sink and cross-modal imbalance. The proposed method tracks positive shifts in visual attention (“gaze shifts”) while processing query tokens to construct a saliency map of task-relevant regions before decoding, and then uses this map to enhance attention to both visual and query tokens during generation. Experiments on multiple models (e.g., LLaVA and Qwen-VL) and hallucination benchmarks (e.g., CHAIR, POPE, MMHal-Bench) show that GIFT reduces hallucination while maintaining general VLM performance with low inference overhead.

**Compliance With Llm Reviewing Policy:**

Affirmed.

**Key Questions For Authors:**

1.The evaluation is conducted on a limited set of LVLMs. Testing on more recent models such as LLaVA-NeXT or InternVL2.5 would strengthen the generality of the conclusions. Since They have a similar architecture to LLaVA-1.5, methods like VCD may also be readily applicable to these models.

2.For contrastive decoding methods, it would be helpful to include comparisons with the SID[1] approach to provide a more comprehensive evaluation.

3.Since the CHAIR metric can be sensitive to the length of generated responses, it would be helpful to report the average generation length or provide additional details about the decoding outputs.

[1]Huo, F., Xu, W., Zhang, Z., Wang, H., Chen, Z., and Zhao, P. Self-introspective decoding: Alleviating hallucinations for large vision-language models. arXiv preprint arXiv:2408.02032, 2024.

**Limitations:**

Yes.

**Strengths And Weaknesses:**

Strengths:

1.The paper is well-written, with a clear and logical organization.

2.The paper identifies two overlooked issues in VLM hallucination mitigation—visual attention sink and imbalanced cross-modal fusion—which are well motivated.

3.The proposed GIFT is an inference-time technique that does not require retraining and introduces only small computational overhead.
4.Experiments across several models and benchmarks demonstrate consistent hallucination reduction while maintaining general VLM performance.

Weaknesses:

1.The method mainly extends existing attention-steering approaches and may be viewed as an incremental improvement.

2.The saliency map relies on query tokens. When queries are weakly related to visual content, the method may produce inaccurate attention guidance.

3.The paper provides a comprehensive evaluation on hallucination benchmarks. However analysis on diverse multimodal reasoning tasks is limited. Including additional evaluations such as GPT-4o–assisted evaluation or MMBench would strengthen the empirical validation and better demonstrate generalization.

---

> ### Author Rebuttal · Authors · 2026-03-27
>
> We thank the reviewer for the positive assessment and the constructive suggestions. We address each point below.
>
> ---
>
> ### W1: Incremental improvement over existing attention-steering approaches
> We appreciate the reviewer's perspective. We agree that GIFT operates within the attention-steering paradigm. We believe the contribution lies in identifying the right signals (gaze shifts over information-rich tokens rather than static attention) and the right balance (joint visual and query enhancement to maintain cross-modal fusion). These insights yield consistent improvements across models and benchmarks with low overhead, and we hope they are useful to the community both as a practical tool and as a basis for future work.
>
> ---
>
> ### W2: Inaccurate guidance when queries are weakly related to visual content
>
> We agree this is a limitation, and we discuss it in Section 6. When the query is not related to the visual content, the resulting saliency map may be noisy. As outlined there, we plan to address this through more selective token identification via LLM prompting or a lightweight classifier that can determine token-level visual relevance. That said, the query-dependent design is a deliberate choice: different queries about the same image should highlight different regions, making the enhancement task-relevant.
>
> ---
>
> ### W3: Limited multimodal reasoning evaluation
> We agree that broader reasoning evaluation would strengthen the paper. Following the reviewer's suggestion, we evaluated GIFT on **MMBench** using the same hyperparameters as other benchmarks:
>
> | Model | Greedy | GIFT |
> |---|---|---|
> | LLaVA-1.5 7B | 73.1 | 73.1 |
> | LLaVA-1.5 13B | 75.6 | 75.8 |
> | Qwen2-VL 7B | 84.6 | 84.6 |
> | Qwen3-VL 8B | 86.9 | 87.2 |
>
> GIFT preserves reasoning performance across all models, demonstrating generalization.
>
> ---
>
> ### Q1: Testing on more recent models
> Thank you for this suggestion. We agree that extending to additional models would further strengthen generality. We prioritized evaluating on recent models with diverse architectures: Qwen2-VL and Qwen3-VL (released in 2024-2025) differ significantly from LLaVA-1.5 in both vision encoder and model design, providing stronger evidence of generalization.
>
> ---
>
> ### Q2: Comparison with SID
> Thank you for this suggestion. We compare GIFT with SID on LLaVA-1.5 7B, where GIFT outperforms SID across all benchmarks:
>
> | Method | CHAIR_s (↓) | CHAIR_i (↓) | POPE F1 (↑) | POPE Acc. (↑) | MMHal Hal. (↓) | MMHal Score (↑) |
> |---|---|---|---|---|---|---|
> | SID | 49.2 | 13.9 | 83.0 | 80.7 | 60.5 | 2.42 |
> | **GIFT** | **39.8** | **10.6** | **83.8** | **81.9** | **57.3** | **2.48** |
>
> GIFT also has lower inference overhead. SID reports 1.35× greedy latency in their paper (668s vs. 494s), while GIFT runs at 1.13× (Table 4). We will include this comparison in the revision.
>
> ### Q3: CHAIR sensitivity to generation length
> This is a fair point. Using the official CHAIR evaluation code, we report average generation lengths (in words) below:
>
> | Model | Greedy | GIFT |
> |---|---|---|
> | LLaVA-1.5 7B | 100.5 | 107.4 |
> | LLaVA-1.5 13B | 100.7 | 105.7 |
> | Qwen2-VL 7B | 98.7 | 100.5 |
> | Qwen3-VL 8B | 402.2 | 447.8 |
>
> GIFT generates comparable or slightly longer outputs across all models, meaning the CHAIR improvements are not driven by generation length. We also note that Qwen3-VL generates more detailed, reasoning-based outputs than other models, which we discuss in Section 5.2 as a factor in its higher CHAIR scores. We will include this table in the revision.
>
> ---
>
> We thank the reviewer for the thoughtful feedback and are happy to discuss any further questions.

---

### Official Review · Reviewer_YTYE · 2026-03-12

**Soundness:** 3
**Presentation:** 3
**Significance:** 4
**Originality:** 3
**Overall Recommendation:** 5
**Confidence:** 4

**Summary:**

This paper addresses hallucination in vision-language models using a training-free inference-time method. The key idea is to build a visual saliency map from positive attention shifts over informative query tokens, rather than from static attention, and then use this signal to steer decoding by jointly enhancing salient visual tokens and calibrating query-token attention. The method is evaluated on several VLMs and benchmarks, and shows consistent hallucination reduction with modest runtime overhead.

**Compliance With Llm Reviewing Policy:**

Affirmed.

**Final Justification:**

The paper is technically sound and empirically strong. It shows consistent gains across multiple models and benchmarks, with solid ablations and reasonable runtime overhead. The main contribution is a meaningful refinement within the inference-time attention steering setting, rather than a fundamentally new framework. My main concerns were the heuristic design choices and the limited grounding of the cross-modal balance argument. The rebuttal addressed these points reasonably well by clarifying the practical rationale and adding further empirical support, including results on an additional benchmark. While the deeper mechanism is still not fully established, this does not change my view that the method is effective and well supported for the scope of the paper.

**Key Questions For Authors:**

1. How sensitive is GIFT to the choice of “information-rich” tokens? Would semantic or learned token selection outperform POS-based filtering?

2. How stable are the selected enhancement layers across datasets and model families?

3. How much of the gain comes from better grounding versus benchmark-specific decoding bias?

**Limitations:**

yes

**Strengths And Weaknesses:**

Strengths:

1. The most novel part is the use of attention shifts rather than static attention to estimate saliency. This is a meaningful refinement over prior inference-time steering heuristics, and the paper provides some evidence that it better localizes task-relevant regions. In particular, the grounding analysis shows higher normalized saliency scores for shift-based maps than static ones.

2. The experimental section is fairly strong. The method is evaluated on multiple architectures and benchmark types, including both hallucination-focused and general VLM benchmarks. The paper also includes ablations on visual enhancement, query calibration, enhancement strength, and layer range, which helps support the proposed design. Runtime overhead is also reported and appears moderate.

Weaknesses:

1. The method remains somewhat heuristic. Several design choices, including POS-based token filtering, layer selection, head filtering, and enhancement ranges, appear empirically chosen rather than derived from a stronger principle. The paper shows that these choices work, but the underlying mechanism is not fully pinned down.

2. The technical novelty is solid but not especially deep. At its core, this is still an inference-time attention manipulation method. The main contribution is a better saliency signal and a more balanced steering rule, rather than a fundamentally new modeling framework.

3. The cross-modal balance argument is intuitively reasonable but not fully established. Scaling query attention in proportion to visual enhancement improves results empirically, but the paper stops short of showing that this truly corresponds to better semantic fusion rather than simply better-tuned decoding dynamics.

---

> ### Author Rebuttal · Authors · 2026-03-26
>
> We sincerely thank the reviewer for the positive assessment and the thoughtful, nuanced feedback. We address the weaknesses and questions below.
>
> ---
>
> ### W1: Heuristic design choices
> We acknowledge that several design choices are empirically motivated rather than theoretically derived. Each has a practical rationale, but we agree that grounding these choices in a stronger theoretical principle would strengthen the work. We consider this an important direction for future research and will clarify the rationale more explicitly in the revision.
>
> ---
>
> ### W2: Technical novelty
>
> We appreciate the reviewer's fair characterization. We agree that GIFT operates within the inference-time attention manipulation paradigm rather than proposing a fundamentally new framework. We believe the contribution lies in identifying the right signals (gaze shifts over information-rich tokens) and the right balance (joint visual and query enhancement) — which together yield consistent improvements across models and benchmarks with low overhead. We hope the practical effectiveness and the insights about visual attention dynamics are valuable to the community.
>
> ---
>
> ### W3: Cross-modal balance argument
>
> We agree that the cross-modal balance mechanism warrants deeper analysis. Our current evidence is empirical: Table 5 shows that GIFT (joint visual + query enhancement) consistently outperforms both "Inc. V." (visual only) and "Cal. V." (calibration only) across all four models. While we cannot definitively isolate whether the gains stem from better semantic fusion or better-tuned decoding dynamics, the consistency across diverse models and benchmarks suggests the benefit is not an artifact of a specific decoding setup. We consider a more rigorous analysis of the fusion mechanism an interesting direction for future work.
>
> ---
>
> ### Q1: Sensitivity to POS-based token selection
>
> We use POS tagging for its simplicity and negligible overhead. We believe semantic or learned selection could further improve results by considering contextual information and assigning per-token relevance weights rather than binary include/exclude decisions, better handling queries that partially or do not depend on visual inputs. We plan to explore this via LLM prompting or a lightweight classifier as discussed in Section 6.
>
> ---
>
> ### Q2: Stability of enhancement layers across datasets and model families
>
> Thank you for this question. Each model requires a different set of enhancement layers, as different architectures process visual information at different depths. For LLaVA-1.5 7B, Table 6 shows that multiple moderate configurations achieve strong performance, indicating the choice is not brittle within a model. We do not yet have a layer ablation for every model, but the selected layers generalize across benchmarks for all four models.
>
> ---
>
> ### Q3: Better grounding versus benchmark-specific decoding bias
> GIFT uses the same hyperparameters across all benchmarks, which span diverse evaluation tasks, i.e. POPE (classification), CHAIR (captioning), MMHal-Bench (open-ended QA). The consistent improvements suggest the gains are not specific to any single evaluation task.
>
> We additionally evaluated GIFT on **AMBER** [1],  a new benchmark not included in the original submission that evaluates hallucination through both discriminative and generative tasks:
>
> | Model | Method | AMBER (↑) | Discriminative (↑) | Generative (↓) |
> |---|---|---|---|---|
> | LLaVA-1.5 7B | Greedy | 83.9 | 74.9 | 7.2 |
> | | **GIFT** | **87.2 (+3.3)** | **80.9 (+6.0)** | **6.6 (-0.6)** |
> | LLaVA-1.5 13B | Greedy | 83.4 | 73.5 | 6.7 |
> | | **GIFT** | **85.7 (+2.3)** | **76.8 (+3.3)** | **5.5 (-1.2)** |
> | Qwen2-VL 7B | Greedy | 84.1 | 74.4 | 6.2 |
> | | **GIFT** | **85.4 (+1.3)** | **76.2 (+1.8)** | **5.5 (-0.7)** |
> | Qwen3-VL 8B | Greedy | 80.2 | 68.3 | 7.9 |
> | | **GIFT** | **83.6 (+3.4)** | **74.0 (+5.7)** | **6.8 (-1.1)** |
>
> GIFT outperforms greedy decoding across all four models, each with the same hyperparameters as other benchmarks, further supporting that the gains are not benchmark-specific.
>
> [1] Wang, Junyang, et al. "Amber: An llm-free multi-dimensional benchmark for mllms hallucination evaluation." arXiv preprint arXiv:2311.07397 (2023).
>
> ---
>
> We thank the reviewer again for the supportive and insightful review, and we are happy to discuss any further questions.

---

> > ### Author Rebuttal · Reviewer_YTYE · 2026-04-03
> >
> > Thanks for careful rebuttal and the additional experiment. I believe the ideas of this work are meaningful after the rebuttal stage. I hope that future work will continue to improve them both technically and theoretically.

---

> > > ### Author Response · Authors · 2026-04-07
> > >
> > > We thank the reviewer for the positive feedback and encouragement. We are committed to incorporating the improvements discussed during the rebuttal into the revised paper, and we look forward to continuing this line of work with both technical and theoretical advancements in future research.

---

### Official Review · Reviewer_55AG · 2026-03-12

**Soundness:** 3
**Presentation:** 3
**Significance:** 3
**Originality:** 3
**Overall Recommendation:** 4
**Confidence:** 4

**Summary:**

This paper proposes GIFT, a training-free hallucination mitigation method for VLMs. The method first computes a saliency map by tracking positive shifts in visual attention over information-rich query tokens, then uses that map during decoding to enhance both visual attention and query attention so that cross-modal fusion remains balanced. The paper belongs to the current line of training-free VLM hallucination mitigation methods that operate at inference time [ref-1, ref-2].

> [ref-1] Li et al., "Evaluating Object Hallucination in Large Vision-Language Models" (EMNLP 2023).
>
> [ref-2] Leng et al., "VCD: Mitigating Object Hallucinations in Large Vision-Language Models via Visual Contrastive Decoding" (CVPR 2024).

**Compliance With Llm Reviewing Policy:**

Affirmed.

**Final Justification:**

The results on a held-out benchmark (W1/Q2) strengthen the generalization case, and W2 and W3 were handled well. POPE and MME remain both tuning targets and primary reported benchmarks, and Q1 (POS heuristic sensitivity) and Q4 (bypass rule for non-visual queries) each got rationale but no experimental follow-up. My recommendation stays at 4.

**Key Questions For Authors:**

Q1. How sensitive is GIFT to the POS-based definition of information-rich query tokens, especially on prompts where the grounding-critical words may not be well captured by that heuristic?

Q2. Can you provide results with hyperparameters chosen without using POPE/MME pseudo-validation, or at least under a frozen tuning protocol selected outside the reported benchmarks?

Q3. Can you add matched inference-time baselines for the Qwen models, or narrow the claim accordingly?

Q4. Given your stated limitation on queries that do not really require visual grounding, did you test a simple gating or bypass rule for such cases?

**Limitations:**

This is a practical paper with a good quality-latency trade-off, but the cleanest version of the result has not quite been shown yet. The main missing piece is a more clearly held-out tuning protocol for the reported benchmarks and a more careful statement of what is actually supported on the Qwen models.

**Strengths And Weaknesses:**

## Strengths

The paper is practically motivated, the method is simple and easy to understand, the quality-latency trade-off is good, and the paper does a nice job checking that hallucination gains are not merely coming from a collapse in general performance.

## Weaknesses

- **[W1] Benchmark-dependent tuning on reported benchmarks**: Appendix D says the enhancement layers are tuned on 10% pseudo-validation splits drawn from POPE and MME, and Figure 6 is then used to choose the enhancement coefficient on those same benchmarks. Since POPE and MME are also part of the reported main results, those results are less clean than they would be under a fully held-out tuning protocol [ref-1].
- **[W2] Overstated cross-model claim**: In Table 2, the Qwen2-VL and Qwen3-VL experiments are only compared against greedy decoding because the other baselines were not reimplemented there. The statement that GIFT outperforms all baselines across models is therefore too broad.
- **[W3] Nontrivial tuning trade-off**: Figure 5 and Table 6 show a real trade-off in which stronger intervention helps hallucination reduction, but overly wide or strong settings start to hurt reasoning. That does not invalidate the method, but it suggests the method is reasonably robust within a moderate range rather than insensitive to tuning.

---

> ### Author Rebuttal · Authors · 2026-03-26
>
> We thank the reviewer for the encouraging assessment and the precise, constructive feedback. We address each concern and question below.
>
> ---
>
> ### W1/Q2: Benchmark-dependent tuning on reported benchmarks
> We appreciate this observation. We followed the pseudo-validation protocol established by Kang et al. (2025) (VAR, one of our baselines), as these hallucination benchmarks lack dedicated validation sets. This ensures a fair comparison under the same tuning setup. We acknowledge the reviewer's point that reporting on the same benchmarks used for tuning is less clean.
>
> To address this, we note that only POPE and MME were used for pseudo-validation tuning. The remaining benchmarks in the paper (CHAIR, MMHal-Bench, and SEED-Bench) are fully held-out evaluations where GIFT shows consistent improvements over greedy decoding. We also provide results on an additional hallucination benchmark, AMBER [1], using the same hyperparameters as other benchmarks, where GIFT consistently outperforms greedy decoding across all models (see table below), demonstrating that the tuned hyperparameters generalize well beyond the tuning benchmarks. We will add AMBER results in the revised paper.
>
> | Model | Method | AMBER (↑) | Discri. (↑) | Gen. Score (↓) |
> |---|---|---|---|---|
> | LLaVA-1.5 7B | Greedy | 83.9 | 74.9 | 7.2 |
> | | **GIFT** | **87.2 (+3.3)** | **80.9 (+6.0)** | **6.6 (-0.6)** |
> | LLaVA-1.5 13B | Greedy | 83.4 | 73.5 | 6.7 |
> | | **GIFT** | **85.7 (+2.3)** | **76.8 (+3.3)** | **5.5 (-1.2)** |
> | Qwen2-VL 7B | Greedy | 84.1 | 74.4 | 6.2 |
> | | **GIFT** | **85.4 (+1.3)** | **76.2 (+1.8)** | **5.5 (-0.7)** |
> | Qwen3-VL 8B | Greedy | 80.2 | 68.3 | 7.9 |
> | | **GIFT** | **83.6 (+3.4)** | **74.0 (+5.7)** | **6.8 (-1.1)** |
>
> ---
>
> ### W2/Q3: Overstated cross-model claim
> We agree this concern is valid. The baseline methods (VAF, Rel-Attn, VAR, VCD) do not provide official implementations for Qwen2-VL or Qwen3-VL, and we chose not to reimplement them to avoid potential discrepancies that could unfairly disadvantage those methods. We will revise the claim to clearly state that GIFT outperforms all baselines on LLaVA models where direct comparisons are available, and consistently improves over greedy decoding across all models including Qwen.
>
> ---
>
> ### W3: Nontrivial tuning trade-off
> We agree with reviewer that both $\alpha$ and the enhancement layer range are robust within moderate settings. Degradation only appears at extreme values, where excessive visual amplification causes the model to over-attend to perceptual details at the expense of reasoning. We will revise the paper to more precisely characterize this robustness.
>
> ---
>
> ### Q1: Sensitivity to POS-based definition of information-rich tokens
> We use POS tagging because it is simple and fast with negligible computational overhead. The tags broadly capture content words, though they may include words that are not visually relevant in certain contexts. Since the saliency map aggregates gaze shifts across all selected tokens and applies min-max normalization, we expect individual noisy tokens to have limited impact. This may become problematic in the two cases discussed in Section 6: (1) the entire query does not require visual inputs, or (2) substantial portions of the query are unrelated to the image content, such as external documents. As outlined in Section 6, we plan to explore more selective token identification via LLM prompting or a lightweight classifier.
>
> ---
>
> ### Q4: Bypass rule for non-visual queries
>
> We have not yet tested a simple gating or a bypass rule for such cases. Determining whether a query requires visual grounding is a semantic and contextual judgment, and many queries partially depend on visual inputs, making a simple rule-based or lexical-based skip decision unreliable. As discussed in Section 6, we plan to explore LLM prompting or a lightweight classifier for token-level relevance scoring, which can better handle the spectrum from fully visual to fully non-visual queries.
>
> ---
>
> We thank the reviewer again for the constructive feedback and are happy to discuss any further questions.
>
> [1] Wang, Junyang, et al. "Amber: An llm-free multi-dimensional benchmark for mllms hallucination evaluation." arXiv preprint arXiv:2311.07397 (2023).

---

> > ### Author Rebuttal · Reviewer_55AG · 2026-04-04
> >
> > W2 and W3 were handled well. The AMBER results are a genuinely useful addition since those hyperparameters were not tuned on that benchmark, and the decision to narrow the Qwen claim is the right call.
> >
> > Two questions are still open for me. On W1/Q2, I appreciate the AMBER evidence, but POPE and MME are still both tuning targets and primary reported benchmarks. CHAIR, MMHal-Bench, and SEED-Bench being held-out helps, but the cleanest fix would be a single frozen hyperparameter setting chosen on a dataset outside the reported suite entirely.
> >
> > On Q1 and Q4, I understand the rationale for POS tagging and the plan to explore lightweight classifiers, but neither has experimental follow-up yet. For Q4 in particular, since non-visual queries are explicitly listed as a failure mode, even a rough estimate of how often this situation arises in the reported benchmarks would be informative.

---

> > > ### Author Response · Authors · 2026-04-07
> > >
> > > We thank the reviewer for the continued constructive feedback and for recognizing our responses to W2 and W3.
> > >
> > > ---
> > >
> > > ### W1/Q2: Frozen hyperparameter setting on external dataset
> > > We agree this is a valid concern. While we followed the pseudo-validation protocol from Kang et al. (2025) for fair baseline comparison, we acknowledge that reporting on tuning benchmarks is less clean.
> > >
> > > To address this, we will report GIFT's performance on the 90% held-out portions of POPE and MME that were excluded from pseudo-validation in the appendix. Combined with the already fully held-out benchmarks (CHAIR, MMHal-Bench, SEED-Bench, and AMBER), this provides a cleaner evaluation of generalization.
> > >
> > > ---
> > >
> > > ### Q1/Q4: POS sensitivity and non-visual queries
> > >
> > > We agree that empirical evidence would strengthen these points. We have conducted experiments using an LLM-based method to select information-rich words as a proxy for the lightweight classifier mentioned in our future work (we currently lack time and data to train a dedicated classifier).
> > >
> > > LLM-based token selection (Q1): We prompted Qwen3-VL 8B with: "Identify all words (including objects, actions, visual attributes, and spatial relations) that require visual information from an image to answer this question. Return them in the order they appear in the question, comma-separated." We evaluated this against our POS-based approach on MMHal-Bench, chosen for its diverse question types:
> > >
> > > | Model | Method | Hal. (↓) | Score (↑) |
> > > |-------|--------|----------|-----------|
> > > | LLaVA-1.5 7B | POS Tag | 57.3 | 2.48 |
> > > |  | LLM-Based | 52.6 | 2.60 |
> > > | Qwen2-VL 7B | POS Tag | 27.5 | 3.58 |
> > > |  | LLM-Based | 26.0 | 3.67 |
> > >
> > > The LLM-based approach selects information-rich words more accurately, leading to better hallucination reduction. We will include this as an option for users prioritizing hallucination mitigation over acceptable latency overhead.
> > >
> > > Non-visual queries (Q4): We expect such cases to be rare in the reported benchmarks, as they are VQA tasks explicitly designed to require visual grounding. For edge cases where queries do not require visual information, the LLM-based solution (or a trained classifier) would handle them more gracefully by identifying when no tokens require visual grounding, allowing the method to bypass saliency computation entirely.
> > >
> > > We will add these results and discussion to the revised paper.

---

### Official Review · Reviewer_Whn7 · 2026-03-13

**Soundness:** 3
**Presentation:** 3
**Significance:** 2
**Originality:** 3
**Overall Recommendation:** 4
**Confidence:** 5

**Summary:**

This paper proposes GIFT, an inference-time method for mitigating VLM hallucination. The key idea is to pre-compute a visual saliency map by tracking positive shifts in visual attention over information-rich query tokens, and then use this saliency map during decoding to enhance attention to salient visual tokens while also proportionally scaling attention to query tokens to preserve cross-modal fusion balance. The paper evaluates GIFT on several hallucination benchmarks and reports consistent improvements over greedy decoding and several baselines, while largely preserving general VLM performance and adding relatively low latency.

**Compliance With Llm Reviewing Policy:**

Affirmed.

**Final Justification:**

I updated the score from 3 to 4 cuz the reviewer has addressed my concern. Here're some details:
- The paper showed several interesting findings and proposed a saliency-map-based method for hallucination mitigation, which is a training-free methods.
- After the rebuttal, the reviewer believes that the method is robust and general enough to handle most VQA tasks across models.
- In terms of those axes, the presentation is good pre-rebuttal, and the soundness/originality becomes good after the discussion; however, the reviewer thinks the significance is still fair cuz the paper doesn't discuss anything about the training-based methods and recently, training-based methods seem to catch up or even surpass training-free mechanisms.

Overall, the reviewer believes the paper is over the acceptance threshold. The reviewer also hopes the authors would add these additional experiments + discussions on training-baed methods to the final/next revision as promised.

**Key Questions For Authors:**

Please see the weakness part.

**Limitations:**

Yes.

**Strengths And Weaknesses:**

**Strengths**

- The paper addressed an important hallucination problem. The attention shit (gaze-shift) idea looks interesting and reasonable.

---

**Weaknesses**

- The method seems to be a bit fragile as they need to choose different layers of the attention and set a specific alpha value. The paper selected the best layer using the "TextVQA" dataset but why choosing that? In addition, Figure 6 suggests that performance is still sensitive to \alpha, and the trade-off is there which the authors have admitted: stronger enhancement can improve hallucination mitigation but also hurt reasoning when over-applied.

- While the authors want to limit the scope in "inference-time" hallucination mitigation. It should at least provide several training-based approaches [3,4,5]. Also, as POPE and CHAIR is almost saturated, the authors should report the results on other open-ended tasks like AMBER and Haloquest [6,7].

- While the gaze shift using saliency map can be a good prior, it's also possible that the questions are asking for background or non-salient regions as the paper mentioned in the limitation as well. It would be great to just show some failure cases.


- (Minor) The paper in essence is to solve the attention sink problem. Have the authors tried any baseline models with registers and it seems like people have found that register or the gated design is enough to solve the attention problem... in a more dundamental way [1, 2].

[1] Darcet, Timothée, et al. "Vision transformers need registers." NeurIPS 2024.

[2] Qiu, Zihan, et al. "Gated attention for large language models: Non-linearity, sparsity, and attention-sink-free." NeurIPS 2025.

[3] Sarkar, Pritam, et al. "Mitigating object hallucination in mllms via data-augmented phrase-level alignment." ICLR 2025.

[4] Wu, Tsung-Han, et al. "Generate, but verify: Reducing hallucination in vision-language models with retrospective resampling." NeurIPS 2025.

[5] Compagnoni, Alberto, et al. "Mitigating hallucinations in multimodal llms via object-aware preference optimization." BMVC 2025.

[6] Wang, Junyang, et al. "Amber: An llm-free multi-dimensional benchmark for mllms hallucination evaluation." arXiv 2023.

[7] Wang, Zhecan, et al. "Haloquest: A visual hallucination dataset for advancing multimodal reasoning." ECCV 2024.

---

> ### Author Rebuttal · Authors · 2026-03-26
>
> We thank the reviewer for the thoughtful feedback and constructive suggestions. We address each concern below and provide new experimental results.
>
> ---
>
> ### W1: Hyperparameter sensitivity
>
> **Layer selection.** We select the layer where positive visual attention shifts peak (Figure 2), as a stronger shift signal provides a clearer distinction between task-relevant and irrelevant visual regions. Different architectures exhibit different dynamics across layers (Figure 2), and the peak layer differs per model. The extraction is a one-time procedure on 50 samples, performed once per model and reused for all tasks. We chose TextVQA for its diversity of visually-grounded questions, but critically, the peak layer is a model-intrinsic property: We repeat the analysis on four additional datasets (MME, POPE, MMHal-Bench, AMBER) with 5 random seeds each, and layer 11 is the peak across datasets and seeds. Due to length limit, we will add the table to the revision.
>
> **Sensitivity to $\alpha$.** We agree that a trade-off exists at extreme values: stronger enhancement improves hallucination mitigation but can hurt reasoning when over-applied. However, GIFT is not sensitive to $\alpha$ within the practical range: $\alpha$ is tuned with a coarse step size of 1.0, and figures 6 show that across moderate values ([1, 4] or [1, 5] depending on the model), hallucination mitigation steadily improves while reasoning performance remains comparable to greedy decoding, i.e. any value in this range yields a net positive outcome with no meaningful trade-off.
>
> ---
>
> ### W2: Training-based mitigation work and additional benchmarks
>
> **Training-based approaches [3,4,5].** We thank the reviewer for pointing out these works. Our Related Work section focused on inference-time methods given the paper's scope, but we agree that discussing recent training-based approaches provides important context. We will add them to the Related Work in the revision and note that GIFT is complementary to these methods, as it can be applied on top of models trained with such techniques for further gains.
>
> **Additional benchmark**: Following the reviewer's suggestion, we evaluated GIFT on AMBER across all four models, using the same hyper-parameters as other datasets:
> | Model | Method | AMBER (↑) | Discriminative (↑) | Generative (↓) |
> |---|---|---|---|---|
> | LLaVA-1.5 7B | Greedy | 83.9 | 74.9 | 7.2 |
> | | **GIFT** | **87.2 (+3.3)** | **80.9 (+6.0)** | **6.6 (-0.6)** |
> | LLaVA-1.5 13B | Greedy | 83.4 | 73.5 | 6.7 |
> | | **GIFT** | **85.7 (+2.3)** | **76.8 (+3.3)** | **5.5 (-1.2)** |
> | Qwen2-VL 7B | Greedy | 84.1 | 74.4 | 6.2 |
> | | **GIFT** | **85.4 (+1.3)** | **76.2 (+1.8)** | **5.5 (-0.7)** |
> | Qwen3-VL 8B | Greedy | 80.2 | 68.3 | 7.9 |
> | | **GIFT** | **83.6 (+3.4)** | **74.0 (+5.7)** | **6.8 (-1.1)** |
>
> GIFT consistently outperforms greedy decoding across all four models, confirming its effectiveness on an additional hallucination benchmark.
>
> ---
>
> ### W3: Failure cases
> We appreciate this concern and would like to clarify that GIFT's visual saliency map is **query-dependent**, not a fixed property of the image. When the query references background regions, gaze shifts naturally follow. For example, on an MMHal-Bench sample with the query "What is the weather like in the image?", GIFT's saliency map correctly focuses on the sky and background (the task-relevant regions rather than foreground objects). This demonstrates that GIFT adapts to the query intent, even for background regions. We will include this example in the revised paper.
>
> That said, as discussed in Section 6, we noticed failure cases when the query is not related to the image or when substantial query portions are unrelated to the image content. We plan to address this by enhancing vision-relevant token identification through LLM prompting or a lightweight classifier, and bypassing saliency computation entirely when no query tokens relate to the image. We will include concrete failure cases in the revised paper.
>
> ---
>
> ### W4: Attention Sink Work
> We thank the reviewer for this suggestion. Registers [1] and gated attention [2] are training-time methods that introduce learnable parameters (extra tokens in ViTs and gating weights per attention head, respectively) to suppress attention sink. In addition to addressing attention sink, GIFT also enhances visual contribution during decoding, which is an aspect these methods do not target. Our ablation (Table 5) illustrates this distinction: GIFT consistently outperforms "Cal. V.", which only recalibrates visual attention to suppress sinks.
>
> We view these approaches as complementary rather than competing: [1,2] can reduce attention sink during training, while GIFT can further reduce sink effects and enhance visual contribution at inference time. The two could be combined for additional benefit. We will add a discussion of [1,2] in the revision.
>
> ---
>
> We thank the reviewer again for the detailed feedback and are happy to discuss any further questions.

---

> > ### Author Rebuttal · Reviewer_Whn7 · 2026-04-01
> >
> > Okay I'm now satisfied with the W2 and W4's response. For W1, I still think the model is parameter-sensitive as we pick TextVQA, a specific dataset, to find the optimal layer and the layer is different across models (L679 - 684: layer 11 for LLaVA-1.5 7B, layer 10 for LLaVA-1.5 13B, layer 14 for Qwen2-VL 7B, and layer 18 for Qwen3-VL 8B --> The layer for different models seem to diverge a lot though...) Can the author debrief this part a bit? Concretely, what datasets are recommended and as the community is focusing on reasoning tasks right now, would the result (i.e., the layer) changes if I chose a math datasets?
> >
> > ---
> >
> > Also for W3, what I mean is that if we asked the model to do image captioning (e.g., "Please describe the image in detail"), how would the system work? As the author mentioned that the method is query-dependent, i cannot imagine what a visual saliency map looks like in this case and how can GIFT improve the model performance (or mitigate the hallucination)?
> >
> > ---
> >
> > That's all my question. I'm happy to discuss these questions with the author further and decide if I wanna change the score later. Thanks!

---

> > > ### Author Response · Authors · 2026-04-02
> > >
> > > Thank you for the follow-up questions!
> > >
> > > ### W1 Extraction layer sensitivity
> > > We appreciate the opportunity to clarify. Different models have varying numbers of layers, training data, and configurations, which naturally leads to different peak layers for positive visual attention shifts. However, the key point is that for each model, the peak layer is consistent across datasets.
> > >
> > > As mentioned in our previous response, we verified this on LLaVA-1.5 7B across TextVQA, MME, POPE, MMHal-Bench, and AMBER with 5 random seeds each - layer 11 consistently peaked across all datasets and seeds.
> > >
> > > To address your question about dataset choice and reasoning tasks: we recommend using datasets where queries require grounding to specific visual content, as they elicit clear attention shifts to task-relevant regions. To demonstrate this with a math reasoning dataset as suggested, we measured the peak layer on MathVista [1], a dataset evaluating math reasoning in visual contexts, for both LLaVA-1.5 7B and Qwen3-VL 8B, each with 5 runs using different seeds. Due to length limits, we show layers surrounding the peak:
> > >
> > > LLaVA-1.5 7B (MathVista):
> > >
> > > | Run | L6 | L7 | L8 | L9 | L10 | L11 | L12 | L13 | L14 | L15 | L16 |
> > > |-----|-----|-----|-----|-----|-----|---------|-----|-----|-----|-----|-----|
> > > | 1 | 0.042 | 0.048 | 0.073 | 0.065 | 0.073 | **0.101** | 0.073 | 0.059 | 0.080 | 0.055 | 0.060 |
> > > | 2 | 0.041 | 0.049 | 0.075 | 0.065 | 0.074 | **0.101** | 0.072 | 0.060 | 0.079 | 0.054 | 0.059 |
> > > | 3 | 0.039 | 0.046 | 0.069 | 0.061 | 0.070 | **0.092** | 0.065 | 0.055 | 0.075 | 0.052 | 0.054 |
> > > | 4 | 0.041 | 0.047 | 0.070 | 0.063 | 0.071 | **0.095** | 0.068 | 0.057 | 0.076 | 0.054 | 0.055 |
> > > | 5 | 0.051 | 0.059 | 0.080 | 0.072 | 0.100 | **0.122** | 0.082 | 0.062 | 0.086 | 0.063 | 0.065 |
> > >
> > > Qwen3-VL 8B (MathVista):
> > >
> > > | Run | L13 | L14 | L15 | L16 | L17 | L18 | L19 | L20 | L21 | L22 | L23 |
> > > |-----|-----|-----|-----|-----|-----|---------|-----|-----|-----|-----|-----|
> > > | 1 | 0.118 | 0.073 | 0.105 | 0.080 | 0.102 | **0.124** | 0.095 | 0.087 | 0.098 | 0.054 | 0.063 |
> > > | 2 | 0.130 | 0.080 | 0.116 | 0.088 | 0.113 | **0.135** | 0.106 | 0.101 | 0.110 | 0.060 | 0.071 |
> > > | 3 | 0.126 | 0.079 | 0.113 | 0.088 | 0.108 | **0.133** | 0.099 | 0.093 | 0.103 | 0.057 | 0.067 |
> > > | 4 | 0.126 | 0.082 | 0.114 | 0.089 | 0.113 | **0.138** | 0.105 | 0.100 | 0.109 | 0.063 | 0.073 |
> > > | 5 | 0.125 | 0.080 | 0.114 | 0.087 | 0.112 | **0.135** | 0.102 | 0.096 | 0.105 | 0.059 | 0.068 |
> > >
> > > The results show that layer 11 for LLaVA-1.5 7B and layer 18 for Qwen3-VL 8B consistently peak across all runs, matching the layers identified on TextVQA. Moreover, the pattern of increases and decreases between consecutive layers is consistent between MathVista and TextVQA (Figure 2). This consistency in both the peak layer and the cross-layer pattern reinforces that the peak layer reflects an intrinsic model property rather than dataset-specific behavior. We will include these MathVista results in the revised paper.
> > >
> > > In summary: the specific layer number differs across models due to architectural differences, but for any given model, the peak layer remains stable across diverse datasets.
> > >
> > > ---
> > >
> > > ### W3 Open-ended Captioning
> > >
> > > This is an excellent question. For captioning queries like "Please describe the image in detail," the query naturally directs attention to generally salient objects and contextually relevant regions throughout the image - similar to how humans would scan the image holistically when describing it, focusing on prominent elements and their spatial relationships. In this case, GIFT's saliency map tends to be more uniformly distributed across multiple informative regions rather than narrowly focused on a single area, reflecting the comprehensive nature of the captioning task.
> > >
> > > GIFT mitigates hallucination in this scenario by suppressing attention sinks on uninformative visual tokens while enhancing attention to genuinely informative regions, preventing the model from inventing details not present in the image. Our results on CHAIR (Table 2), which evaluates hallucination in open-ended image captioning, demonstrate that GIFT consistently improves performance on this type of task. Notably, GIFT generates comparable or slightly longer captions (please check Q3 for Reviewer 5Rxy), meaning the CHAIR improvements are not driven by shorter outputs but by more accurate visual grounding. For example, on a CHAIR sample showing a man playing tennis, GIFT's saliency map appropriately focuses on the person, the tennis ball, and the playfield - the key elements needed for comprehensive image description. We will include additional examples with visualizations of the saliency maps for captioning queries in the revision.
> > >
> > > ---
> > >
> > > We hope this addresses your concerns and are happy to discuss further. Thank you again for the constructive engagement!
> > >
> > > References:
> > >
> > > [1] Lu, Pan, et al. "Mathvista: Evaluating mathematical reasoning of foundation models in visual contexts." arXiv preprint arXiv:2310.02255 (2023).

---

### Decision · Program_Chairs · 2026-04-30

**Decision:**

Accept (regular)

**Comment:**

This paper proposes a technically solid, training-free inference-time method that effectively mitigates VLM hallucination with modest latency, demonstrating consistent improvements across models and benchmarks. The authors have also responsively addressed many reviewer concerns, resulting in a consistent positive recommendation from all reviewers. However, unresolved issues—including non-rigorous hyperparameter tuning on reported benchmarks, insufficient handling of non-visual queries, limited theoretical grounding for heuristic design choices, and a narrow evaluation scope—limit its impact, and the authors are expected to address their committed revisions in the final manuscript.